# OptiBench Meets ReSocratic: Measure and Improve LLMs for Optimization Modeling

**Zhicheng Yang[1]  Yiwei Wang[3]  Yinya Huang[4]  Zhijiang Guo[1]  Wei Shi[6]**
**Xiongwei Han[6]  Liang Feng[9]  Linqi Song[5]  Xiaodan Liang[7,8]  Jing Tang[1,2]***
[1]The Hong Kong University of Science and Technology (Guangzhou)
[2]The Hong Kong University of Science and Technology   [3]University of California, Merced
[4]ETH Zurich   [5]City University of Hong Kong   [6]Huawei Noah's Ark Lab
[7]Sun Yat-sen University   [8]MBZUAI   [9]Chongqing University
{yangzhch6, wangyw.evan, xdliang328}@gmail.com,
yinya.huang@hotmail.com, zhijiangguo@hkust-gz.edu.cn,
jingtang@ust.hk

## ABSTRACT

Large language models (LLMs) have exhibited their problem-solving abilities in mathematical reasoning. Solving realistic optimization (OPT) problems in application scenarios requires advanced and applied mathematics ability. However, current OPT benchmarks that merely solve linear programming are far from complex realistic situations. In this work, we propose OptiBench, a benchmark for end-to-end optimization problem-solving with human-readable inputs and outputs. OptiBench contains rich optimization problems, including linear and nonlinear programming with or without tabular data, which can comprehensively evaluate LLMs' solving ability. In our benchmark, LLMs are required to call a code solver to provide precise numerical answers. Furthermore, to alleviate the data scarcity for optimization problems, and to bridge the gap between open-source LLMs on a small scale (e.g., Llama-3-8b) and closed-source LLMs (e.g., GPT-4), we further propose a data synthesis method namely **ReSocratic**. Unlike general data synthesis methods that proceed from questions to answers, **ReSocratic** first incrementally synthesizes formatted optimization demonstrations with mathematical formulations step by step and then back-translates the generated demonstrations into questions. Based on this, we synthesize the ReSocratic-29k dataset. We further conduct supervised fine-tuning with ReSocratic-29k on multiple open-source models. Experimental results show that ReSocratic-29k significantly improves the performance of open-source models.

## 1 INTRODUCTION

Large language models (LLMs), such as GPT-3 (Brown et al., 2020), GPT-4 (Achiam et al., 2023), and Llama (Touvron et al., 2023a;b), have demonstrated their superior capability in logical reasoning (Suzgun et al., 2023; Huang et al., 2023) and mathematical reasoning (Ling et al., 2017; Patel et al., 2021; Yang et al., 2022; 2023), such as solving elementary (Cobbe et al., 2021) to high-school level (Hendrycks et al., 2021) math problems. Yet a follow-up curiosity is to what extent LLMs apply their mathematical intelligence to practical scenarios. Optimization problem solving (Ramamonjison et al., 2022b; Xiao et al., 2023; AhmadiTeshnizi et al., 2024; Huang et al., 2024a) is a field of applied mathematics that has been proven beneficial in many applications such as supply chain management, power energy scheduling, marketing, and quantitative trading. Optimization problem-solving is a comprehensive task that evaluates the mathematical and coding capabilities of LLMs. To provide the optimal solution to an optimization problem, LLMs are not only required to understand and construct the mathematical formulation according to the given problem but also to call an optimization solver to get the final answers.

---

*Corresponding Author: Jing Tang (jingtang@ust.hk).

Table 1: Comparison of optimization problem solving benchmarks. "End2End" indicates whether the benchmark requires the model to solve for the optimal values of the variables and the optimization objective. "Implicit/Explicit" refers to whether the numeric values in the question are displayed.

| Benchmark | Question Form | Size | End2End | Linear | | Nonlinear | |
|---|---|---|---|---|---|---|---|
| | | | | w/ table | w/o table | w/ table | w/o table |
| ComplexOR (Xiao et al., 2023) | Implicit | 37 | √ | × | √ | × | × |
| NLP4LP (AhmadiTeshnizi et al., 2024) | Implicit | 57 | √ | × | √ | × | × |
| NL4OPT (Ramamonjison et al., 2022b) | Explicit | 289 | × | × | √ | × | × |
| OPTIBENCH (**Ours**) | Explicit | **605** | √ | √ | √ | √ | √ |

Previous studies (Ramamonjison et al., 2022a; Xiao et al., 2023; AhmadiTeshnizi et al., 2024) have explored using LLMs to solve operations research problems. However, these studies have not yet extended to more generalized scenarios regarding practical optimization problems. Specifically, NL4OPT (Ramamonjison et al., 2022a;b) uses named entity recognition to extract entity and numerical values in the given question text, and then formulate it into mathematical models. They only measure the model's ability to correctly construct mathematical formulations, without considering solving the mathematical formulations being constructed. To further evaluate models providing the final optimal solution, i.e., the numerical values of the variables and the optimization objective, ComplexOR (Xiao et al., 2023) and NLP4LP (AhmadiTeshnizi et al., 2024) benchmark the models to solve a problem with an optimization solver in the setting without explicit input numbers. However, due to the difficulty of collecting such data, these benchmarks are still on a small scale. Moreover, the recent MAMO (Huang et al., 2024a) proposes to further benchmark optimization problem solving with a code solver. Nevertheless, one common pitfall of all the aforementioned works is that they merely focus on linear programming, whereas nonlinear optimization problems and practical tabular format are not included. Table 1 provides a comparison of the aforementioned benchmarks. We also discuss the differences between current benchmarks in detail in Appendix A.

To bridge this gap, we propose OPTIBENCH, a new benchmark with high-quality data to evaluate LLMs' end-to-end solving ability in optimization tasks. We carefully select 605 questions and conduct careful manual verification to form the dataset. OPTIBENCH contains linear and nonlinear programming with both integer and mixed integer variables in the programming problems. OPTIBENCH also includes tabular data, which fills the gap in current optimization benchmarks. Figure 1 demonstrates OPTIBENCH examples in the four problem types. A model solves an OPTIBENCH problem by reading the natural language input and then generating Python code that solves the problem, where the code will be processed to acquire the numerical value of the variables and the objective function.

Additionally, the data scarcity issue in optimization tasks (Xiao et al., 2023; AhmadiTeshnizi et al., 2024) cannot be ignored. Annotators are required to possess good professional knowledge, making the process not only expensive but also time-consuming and labor-intensive. In addition, there is a significant performance gap between small open-source models (e.g., Llama-2-7B, Llama-3-8B) and large models (e.g., GPT-4) in many complex reasoning tasks (Ling et al., 2017; Patel et al., 2021; Suzgun et al., 2023; Yang et al., 2022; 2023; Huang et al., 2023). To this end, we propose **ReSocratic**, a novel method for synthesizing diverse and reliable data for optimization problems. Unlike previous methods that synthesize questions first and then answers, **ReSocratic** incrementally synthesizes the formatted optimization demonstration in a reverse manner, and finally back-translates it into a question. Benefiting from intermediate reasoning steps, the quality of **ReSocratic**'s synthetic data is higher than that of previous methods. We collect 29k samples with **ReSocratic**, resulting in the RESOCRATIC-29K dataset. In summary, our contributions are as follows:

- We introduce a high-quality benchmark OPTIBENCH for optimization problems with complex instances in multiple forms. As far as we know, this is the first large-scale benchmark including nonlinear and tabular data to measure LLMs' end-to-end problem solving abilities. We conducted an in-depth evaluation of a range of LLMs under various settings.

- We propose **ReSocratic**, a novel method for generating diverse and reliable data for optimization problems. In particular, **ReSocratic** synthesizes complex reasoning data from scratch in a reverse manner.

- We synthesize the RESOCRATIC-29K dataset with 29k samples by using our **ReSocratic**. Experimental results show that conducting supervised fine-tuning with RESOCRATIC-29K significantly improves the performance of open-source models on OPTIBENCH (e.g., Llama-2-7B-Chat from 0.0% to 30.6%; Llama-3-8B-Instruct from 13.6% to 51.7%), which further demonstrates the validity of our synthetic data.

---

### linear problem without table

**Textual Question:**

There are two ways to extract a metal from mined ores. The first way is to use process J and the second is process P. Process J can extract 5 units of metal using 8 units of water and produces 3 units of pollution. Process P can extract 9 units of metal using 6 units of water and produces 5 units of pollution. There can be at most 1500 units of water 1350 units of pollution. How many of each type of processes should be performed to maximize the amount of metal extracted?

*Optimization Model:*

Define $j$ *as* the process number in the type of J, $p$ as the process number in the type of P.

**Optimization Target:**

$$\max_{j,p \in N} \quad 5j + 9p$$

**Constraints:**

$$s.t. \quad \begin{aligned} 8j + 6p &\leq 1500 \\ 3j + 5p &\leq 1300 \end{aligned}$$

---

### nonlinear problem without table

**Textual Question:**

A piece of cardboard is 1 meter by 1/2 meter. A square is to be cut from each corner and the sides folded up to make an open-top box. What are the dimensions of the box with maximum possible volume?

*Optimization Model:*

Define $W,L$ *as* the width and length of the cardboard respectively, $x$ as the length of each corner to be cut to make the box.

**Optimization Target:**

$$\max_{x} \quad (W - 2x)(L - 2x)x$$

**Constraints:**

$$s.t. \quad \begin{aligned} 2x &\leq W \\ 2x &\leq L \end{aligned}$$

---

### linear problem with table

**Textual Question:**

There are six cities (cities 1-6) in Kilroy County. The county must determine where to build fire stations. The county wants to build the minimum number of fire stations needed to ensure that at least one fire station is within 15 minutes (driving time) of each city. The times (in minutes) required to drive between the cities in Kilroy County are shown in the following Table.
Tell Kilroy how many fire stations should be built and where they should be located.

Table (Time Required to Travel between Cities in Kilroy):

| From / To | City 1 | City 2 | City 3 | City 4 | City 5 | City 6 |
|---|---|---|---|---|---|---|
| City 1 | 0 | 10 | 20 | 30 | 30 | 20 |
| City 2 | 10 | 0 | 25 | 35 | 20 | 10 |
| City 3 | 20 | 25 | 0 | 15 | 30 | 20 |
| City 4 | 30 | 35 | 15 | 0 | 15 | 25 |
| City 5 | 30 | 20 | 30 | 15 | 0 | 14 |
| City 6 | 20 | 10 | 20 | 25 | 14 | 0 |

*Optimization Model:*

Define $x_i$ ($i = 1,2,...,6$) as whether the station should be build in City $i$.

**Optimization Target:**

$$\min_{x_1,x_2,\cdots,x_6 \in B} \sum_{i=1}^{6} x_i$$

**Constraints:**

$$s.t. \quad \begin{aligned} x_1 + x_2 &\geq 1 \\ x_1 + x_2 + x_6 &\geq 1 \\ x_3 + x_4 &\geq 1 \\ x_3 + x_4 + x_5 &\geq 1 \\ x_4 + x_5 + x_6 &\geq 1 \\ x_2 + x_5 + x_6 &\geq 1 \end{aligned}$$

---

### nonlinear problem with table

**Textual Question:**

A company is planning to optimize its production of five different products (Product A, Product B, Product C, Product D, and Product E) to maximize profit while considering the environmental impact of production. The profit per unit and the environmental impact per unit for each product are given in the following Table.

| Product | Profit per Unit | Environmental Impact per Unit |
|---|---|---|
| A | $50 | 10 units |
| B | $70 | 15 units |
| C | $60 | 12 units |
| D | $80 | 20 units |
| E | $90 | 18 units |

The company has a total production capacity of 1500 units across all products. The company must produce at least 200 units of Product A and 300 units of Product B to fulfill contractual obligations. The total environmental impact should not exceed 20,000 units. The company wants to maximize the Profit-Impact ratio, where the Profit-Impact ratio is defined as the total profit divided by the total environmental impact.

*Optimization Model:*

Define $a - e$ as the number of products of A-E that the company should produce.

**Optimization Target:**

$$\max_{a,b,c,d,e \in N} \quad P/I$$

**Constraints:**

$$\begin{aligned} P &= 50a + 70b + 60c + 80d + 90e \\ I &= 10a + 15b + 12c + 20d + 18e \\ s.t. \quad a + b + c + d + e &\leq 5000 \\ a &\geq 200 \\ b &\geq 300 \\ 10a + 15b + 12c + 20d + 18e &\leq 2000 \end{aligned}$$

Figure 1: Our OPTIBENCH contains various types of data (linear, nonlinear, table). To enhance readability, we present the table in an Excel format and include a diagram to illustrate the nonlinear example without a table.

## 2 RELATED WORK

**Benchmarks for Optimization Modeling.** More closely related to our approach, the NL4OPT benchmark (Ramamonjison et al., 2022a;b) investigates controlled generation techniques to obtain an automatic suggestion of formulations. They first use named entity recognition methods to extract a set of entity-typed declarations, then they transform it into linear program models. As one can see, NL4OPT only evaluates an AI model's ability to establish mathematical models, while we

contribute an end-to-end framework in this work. Optimus (AhmadiTeshnizi et al., 2024) and ComplexOR (Xiao et al., 2023) also make significant research in the field of operations research with LLMs. However, they provide a minimal test set, containing less than 70 test samples. Recently, MAMO (Huang et al., 2024a) is proposed to benchmark mathematical modeling with code solvers. However, all these works merely focus on linear programming, ignoring the nonlinear problems that exist widely in practical applications. In addition, these benchmarks are simple in form, ignoring the tabular data that often occurs in industrial scenarios. Additionally, we notice a work Tang et al. (2024) explores synthesizing problems via a semi-automated process. This work is based on forward synthesis. Moreover, they did not focus on tabular data and nonlinear problems in real scenarios. In contrast, in this paper, we aim to benchmark practical optimization modeling with a high-quality manually checked test-bed OPTIBENCH and also automatically synthesize more comprehensive optimization data including tabular data and code solutions resulting in RESOCRATIC-29K. In this work, we contribute OPTIBENCH, which is an end-to-end benchmark containing 605 multi-type data samples. OPTIBENCH is a comprehensive benchmark that involves linear, non-linear, and tabular data, and the types of variables involved in the problems include continuous, integers (IP), and mixed integers (MIP).

**Data Synthesis for Mathematical Reasoning.** Improving the performance of language models in mathematical reasoning tasks significantly depends on increasing the quantity of fine-tuning data for LLMs. A substantial body of work has been dedicated to these areas (Yu et al., 2023; Liu & Yao, 2024; Li et al., 2023; Yuan et al., 2023; Lu et al., 2024b; Yue et al., 2023). One notable approach is Rejection Sampling Fine-Tuning (RFT; Yuan et al. 2023), which employs supervised models to generate and collect correct reasoning paths, creating augmented fine-tuning datasets. MAmmoTH (Yue et al., 2023) utilizes RFT with GPT-4 to gather both Chain-of-Thought solutions in natural language and Program-of-Thought solutions in formal language. Similarly, MetaMath (Yu et al., 2023) focuses on data augmentation for both question and answer texts. MathGenie (Lu et al., 2024b) collects a vast amount of data through open-source language models. While there has been significant progress in synthesizing data for informal mathematical reasoning, efforts have also been made to address formal reasoning through the use of formal languages and compilers (Xiong et al., 2023; Huang et al., 2024b; Lu et al., 2024a). However, a major challenge remains: there is a scarcity of high-quality data for optimization problems. This lack of data limits the direct application of these prior approaches to optimization contexts.

**Socratic Method.** The Socratic method (Gose, 2009; Scholle, 2020) is a critical thinking method with dialogic disassembled multi-step sub-questions and answers cultivating in answering a complex question. This method has been applied by current language model techniques for advanced reasoning tasks, such as prompting step-wise reasoning (Qi et al., 2023; Chang, 2023; Shridhar et al., 2022), multi-agent interaction (Zeng et al., 2023), and discovering math knowledge (Dong et al., 2023). For example, Qi et al. (2023) proposes a divide-and-conquer style algorithm that mimics recursive thinking by asking Socratic questions, it thus relieves the reliance on the initial decision as chain-of-thought (CoT) and achieves performance improvements on several complex reasoning tasks. Chen & Lampouras (2023); Xie et al. (2023) utilize back-translation as core modules in their data generation frameworks, but our ReSocratic places more emphasis on step-by-step reverse construction of the chain-of-thought from scratch, with back-translation being one tiny step in our framework. Another line of work (Ang et al., 2023; Cobbe et al., 2021) applies the Socratic method for fine-grained dataset construction. GSM8K Socratic dataset[1] (Cobbe et al., 2021) is the most related work to our paper. They inject automatically generated "Socratic sub-questions" before each step, resulting in fine-grained math data. To construct a step-by-step benchmark for optimization problem solving with intermediate solutions, in this work, we explore the Socratic method to synthesize optimization problems. Unlike the previous study, we propose a reverse Socratic approach (**ReSocratic**) that generates optimization problems from the answer back to a question, and we demonstrate its superiority to traditional forward Socratic synthesis.

## 3 OPTIBENCH: BENCHMARK FOR OPTIMIZATION MODELING

The benchmark OPTIBENCH is to evaluate the capability of large language models to solve end-to-end optimization problems. Table 1 compares OPTIBENCH and related optimization-problem

---

[1] https://github.com/openai/grade-school-math?tab=readme-ov-file#socratic-dataset

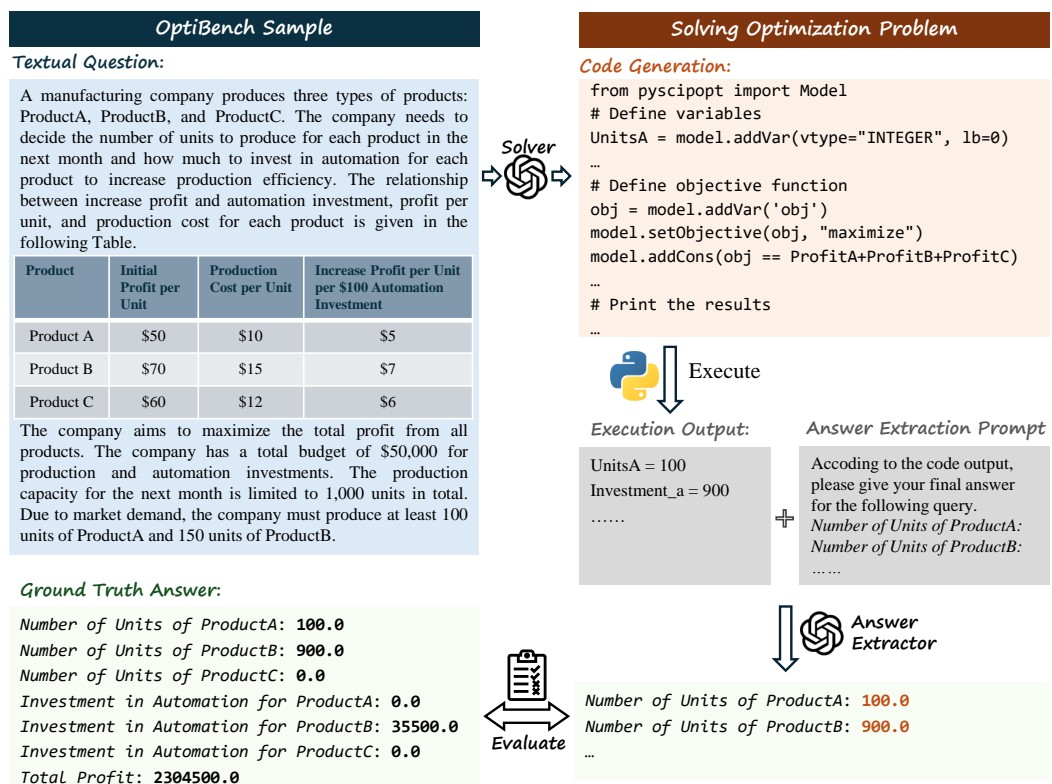

Figure 2: Evaluation procedure of an OPTIBENCH example. This example is about a mixed integer nonlinear optimization problem. The LLM is first required to write code to solve the question. Then, the LLM is required to extract the exact numbers according to the code execution output.

benchmarks. OPTIBENCH covers a substantial number of optimization problems with a wider range of problem types. Specifically, OPTIBENCH features linear programming (linear), non-linear optimization problems (non-linear), and table content as in industrial use (Table), resulting in a comprehensive and versatile benchmark for LLM optimization problem-solving. OPTIBENCH is an end-to-end benchmark, that takes natural language as input and numerical values of variables and objective as output. We show the four types of questions in Figure 1.

**Data Collection and Annotation.** In the data annotation stage, we assign workers to collect questions from textbooks (Bertsimas & Tsitsiklis, 1997; Conforti et al., 2014; Wolsey, 2020), and a university's course assignments and examinations. We require our workers to write Python code, call the pyscipopt[2] solver to solve each problem, and ask them to output the values of the variables and optimization targets at the end of the code. Figure 2 shows a mixed integer nonlinear programming sample of OPTIBENCH. For each sample, we provide the "**Question**" and "**Results**". More details of data collection and annotation are shown in Appendix D.

**Data Statistics.** In Figure 4, we show the statistical results of four data formats (linear w/ table, linear w/o table, nonlinear w/ table, and nonlinear w/o table). Overall, our OPTIBENCH exhibits good diversity in terms of question types, number of variables, and text length.

**Evaluation.** Unlike NL4OPT (Ramamonjison et al., 2022b), which only measures the mathematical modeling ability of the language model, we also measure the solving ability of the language model to call code solver. In this paper, the evaluation approach we adopt is an end-to-end process where natural language text is the input and numerical form answers are the output. Given an optimization problem $p$, the LLM is required to generate a solution including code $c = \text{LLM}(p)$. Next, a Python interpreter is used to execute the code and obtain the code output $o = \text{Python}(c)$. Then, we request the LLM to give the numerical form answer $a_i = \text{LLM}([o, r_i])$ for each variable and objective in the problem, where $r_i$ is the natural language description of "**Ground Truth Answer**" in Figure 2.

---

[2]https://github.com/scipopt/PySCIPOpt

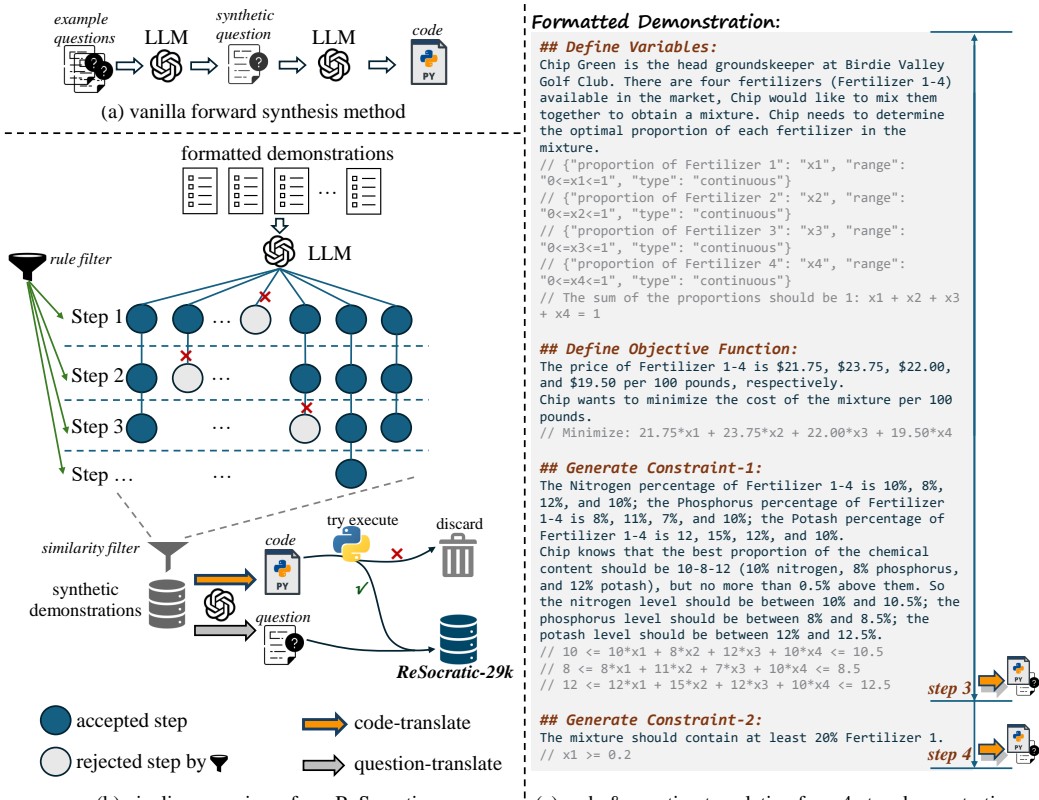

Figure 3: (a) The forward data synthesis method is to synthesize the question first, and then let the LLM generate the answer to the synthetic question. (b) In contrast, **ReSocratic** first synthesizes carefully designed formatted demonstration and then transforms it into code and questions. (c) An example of a formatted demonstration.

Finally, we compare the numerical form answer $a_i$ with the ground truth answer $a_i^*$ to calculate the accuracy. A problem is considered solved if and only if all the variables and objectives are correctly matched.

## 4 ReSocratic: Data Synthesis with Reverse Socratic

In this section, we introduce **ReSocratic**, a novel data synthesis method for eliciting diverse and reliable data. The **ReSocratic** framework is shown in Figure 3(a). Former methods (forward synthesis) skipped the intermediate reasoning steps and directly generated the question, relying more on the intuition of LLMs. Whereas, the main idea of **ReSocratic** is to incrementally synthesize an optimization problem with step-by-step generation via the Socratic method in a reverse manner from our elaborately formatted seed demonstrations to questions. Our **ReSocratic** consists of three steps: 1) Seed Demonstration Formalization, 2) New Demonstration Synthesis, and 3) Question and Code Translation.

**1) Seed Demonstration Formalization**. The seed demonstrations are rigorously selected by humans and each seed data is of diverse operations research scenarios. Figure 3(b) shows an example of seed demonstration. It is formatted step by step, where each step is clearly delineated and builds upon the previous one. Each step consists of three parts:1) A header with "##" that introduces the specific aspect of the demonstration being addressed, such as "*Defining Variables*", "*Objective Function*", or "*Constraints*". 2) A narrative description (colored in blue) in natural language that provides context and details about the element being introduced. This helps to understand the rationale and the requirements of that particular part of the optimization problem. 3) Mathematical formalization following "//" that translates the natural language description into a precise mathematical expression or constraint.

**2) New Demonstration Synthesis**. The ReSocratic method prompts an LLM to generate new demonstrations based on the seeds. We sample 2 seeds each time from the pool to form the synthesis prompt as shown in Appendix E.2. The LLM will follow the given prompt to generate new demonstrations step by step. The generated data is rigorously selected via (1) each intermediate step should follow the format (as shown in the first step) by a *rule filter* and (2) the overall demonstration does not overlap with generated ones by a *similarity filter*. For all generated demonstrations, we set a *similarity filter*, which converts all texts into TF-IDF vectors and filters out demonstrations with cosine similarity higher than a threshold. The procedure is shown in Figure 3(b).

**3) Question and Code Translation**. We construct a code generation prompt to solve the mathematical formulations in the synthetic demonstrations and output the optimal solution results. If the code runs incorrectly, we delete this demonstration. Next, to acquire the questions in plain text format and the questions in table format, we construct two back-translation prompts. All the prompts are shown in Appendix E. Then, for each generated demonstration starting from the third step, we translate it into a question-code pair, as shown in Figure 3(b). Each generated code will be executed automatically to ensure there are no bugs.

## 5 EXPERIMENTS AND ANALYSIS

### 5.1 BASELINES AND SETTINGS

**Evaluation Setting.** The evaluation metric of our OPTIBENCH is the answer accuracy, as detailed in Section 3. We show the solving accuracy of the four data types along with the code pass rate. We evaluate LLMs under three settings: Zero-shot, Few-shot, and Supervised Fine-Tuning (SFT) setting. We provide the zero-shot prompt and the few-shot prompt to solve the problem in Appendix E.1.

**Baselines.** We select **GPT-family** (Brown et al., 2020; Achiam et al., 2023), **Llama-family** (Touvron et al., 2023b; Team, 2024), Qwen2 Yang et al. (2024), Mistral-v0.3 Jiang et al. (2023), and **DeepSeek-family** (DeepSeek-AI, 2024) as the baselines in zero-shot and few-shot settings. For the SFT setting, we use **Llama-2-7B-Chat** and textbfLlama-3-8B-Instruct.

**Setting of Data Synthesize**. We use DeepSeek-V2 (DeepSeek-AI, 2024) to apply **ReSocratic**. As an open-source large language model, DeepSeek-V2 (DeepSeek-AI, 2024) stands out due to its competitive performance to GPT-4, while concurrently offering a more cost-effective alternative. Furthermore, it exhibits a superior throughput, approximately 6 times greater, when contrasted against the existing 70b open-source model (DeepSeek-AI, 2024). Utilizing the advanced capabilities of DeepSeek-V2, we contribute 29k synthetic data. This results in the RESOCRATIC-29K dataset. The threshold of the aforementioned *similarity filter* is set at 0.7, we also set the *temperature* as 0.7, and sample 50 responses for each query.

**Fine-tuning Setting**. For a given language model, we utilize our contributed RESOCRATIC-29K to conduct supervised fine-tuning. Based on this, we conduct fine-tuning experiments on two A800 GPUs, the epoch is set as 3, the learning rate is $2e^{-5}$, and the batch size is 128.

### 5.2 DATA STATISTICS AND VISUALIZATION

For both OPTIBENCH and RESOCRATIC-29K, we show the statistical results of data distribution in question type, variable numbers, and question length. The question length refers to the number of characters in the question text. The results are shown in the following Figure 4. The distribution of variable numbers in both OPTIBENCH and RESOCRATIC-29K generally conforms to the long-tail distribution. In addition, the distribution of question length in OPTIBENCH also conforms to the long-tail distribution, while RESOCRATIC-29K is more balanced.

Furthermore, we show the visualization results of OPTIBENCH and RESOCRATIC-29K using the t-SNE algorithm based on question semantic embedding. The visualization results of each type (linear w/o table, linear w/ table, nonlinear w/o table, nonlinear w/ table) are shown in Figure 5.

In general, from the statistical results and visualization results, our RESOCRATIC-29K has a good diversity in the question types, variable numbers, question length, and text semantics.

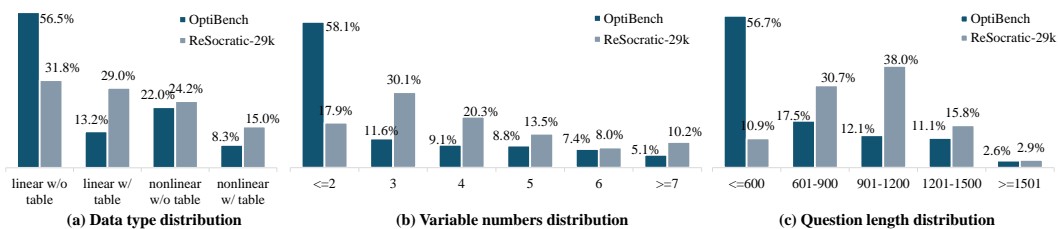

Figure 4: Statistical results of OPTIBENCH and RESOCRATIC-29K

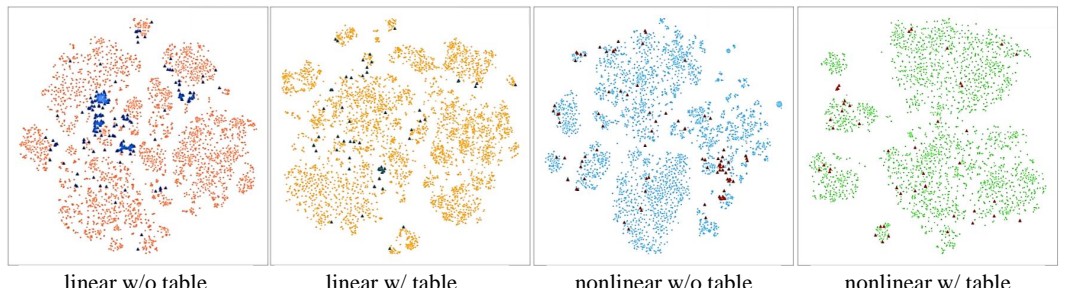

| linear w/o table | linear w/ table | nonlinear w/o table | nonlinear w/ table |

Figure 5: t-SNE visualization results of OPTIBENCH and RESOCRATIC-29K. ('△' indicates the data point of OPTIBENCH and '•' indicates the data point of RESOCRATIC-29K)

## 5.3 MAIN RESULTS

As shown in Table 2, GPT-4 has the strongest overall performance and achieves state-of-the-art performance on almost all kinds of data formats. The performance of the two open-source models, llama3-70b and deepseek-v2, is close to that of GPT-4 in the few-shot setup. In addition, open source small models perform extremely poorly on OPTIBENCH, with Llama-2-7B-Chat not getting a single question correctly solved and Llama-3-8B-Instruct getting only 13.6% accuracy on the few-shot setting. From the perspective of data type, the nonlinear data of our OPTIBENCH is more challenging than the linear data, and the data with table (w/ table) is more challenging than without table (w/o table). Then, to show the validity of **ReSocratic**, we SFT Llama-2-7B-Chat, and Llama-3-8B-Instruct with our synthetic data RESOCRATIC-29K. We improved the performance of the Llama-2-7B-Chat from 0.0% to 30.6%, and the Llama-3-8B-Instruct from 13.6% to 51.1% (+37.5%), which is very close to the GPT-3.5-Turbo. In addition, Llama-3-8B-Instruct even exceeds GPT-4 in the data type of linear w/table, reaching state-of-the-art performance. We present a more detailed dataset performance analysis in Figure 6.

## 5.4 PERFORMANCE ANALYSIS ON DATA SPLIT

We show the detailed evaluation results under the few-shot setting for GPT-4 and GPT-3.5-Turbo in Figure 6. The performance on OPTIBENCH is split by data type, variable numbers, and question length. According to the results, we can find that:

**1) Tabular questions are harder.** It is more difficult to solve table questions than no-table questions, and nonlinear questions are more difficult than linear questions.

**2) Nonlinear problem is harder.** According to Table 2, for all language models, the performance of nonlinear questions is significantly lower than that of linear questions. In Figure 6 (a), we show the average performance of linear problems and nonlinear problems for GPT-4 and GPT-3.5-Turbo. The performance is 66.9% for linear problems and 30.8% for nonlinear problems.

**3) In general, questions with more variables and longer text lengths are more difficult.** We show the performance of GPT-4 and GPT-3.5-Turbo on the data with different numbers of variables in Figure 6 (b). From the linear trend line we provided, we can see that model performance decreases as the number of variables increases. As can be seen from Figure 6 (c), GPT-3.5-Turbo has a significant performance degradation on long text questions, while GPT-4 has a more balanced performance on different text lengths.

Table 2: Main results on OPTIBENCH. "Code Pass" refers to the success rate of code execution. **Bold** indicates the sota in the current setting, underline indicates the sota in the overall setting.

| Model | Linear | | Nonlinear | | All | Code Pass |
|---|---|---|---|---|---|---|
| | w/o Table | w/ Table | w/o Table | w/ Table | | |
| *Zero-shot Prompt* | | | | | | |
| Llama-3-8B-Instruct | 0.0% | 0.29% | 0.0% | 0.0% | 0.17% | 8.8% |
| Llama-3-70B-Instruct | **76.9%** | 50.0% | 30.8% | 32.0% | 59.5% | 86.8% |
| Mistral-7B-Instruct-v0.3 | 0.6% | 0.0% | 0.0% | 0.0% | 0.3% | 6.9% |
| Qwen2-7b-Instruct | 3.5% | 0.0% | 3.0% | 0.0% | 2.6% | 19.2% |
| DeepSeek-V2 | 40.4% | 27.5% | 29.3% | 18.0% | 34.4% | 74.0% |
| DeepSeek-V2.5 | 78.4% | **67.5%** | 33.1% | 24.0% | 62.5% | 92.7% |
| GPT-3.5-Turbo | 68.1% | 37.5% | 19.5% | 16.0% | 49.1% | 85.0% |
| GPT-4 | 75.4% | 62.5% | 42.1% | 32.0% | 62.8% | 88.8% |
| GPT-4o-mini | 76.0% | 48.8% | 35.3% | 34.0% | 60.0% | 84.8% |
| GPT-4o | **78.1%** | 65.0% | **45.9%** | **40.0%** | **66.1%** | **90.1%** |
| *Few-shot Prompt* | | | | | | |
| Llama-3-8B-Instruct | 17.8% | 2.5% | 11.3% | 8.0% | 13.6% | 26.9% |
| Llama-3-70B-Instruct | 79.2% | 57.5% | 33.8% | 32.0% | 62.5% | 91.2% |
| Mistral-7B-Instruct-v0.3 | 40.0% | 23.8% | 13.5% | 18.0% | 27.9% | 83.8% |
| Qwen2-7b-Instruct | 65.5% | 27.5% | 18.8% | 14.0% | 46.0% | 87.6% |
| DeepSeek-V2 | 79.5% | 56.3% | 27.1% | 32.0% | 61.0% | 85.5% |
| DeepSeek-V2.5 | 79.5% | **71.3%** | 40.6% | 48.0% | 67.3% | 91.2% |
| GPT-3.5-Turbo | 75.4% | 40.0% | 28.6% | 26.0% | 56.4% | **93.2%** |
| GPT-4 | 80.7% | **71.3%** | 34.6% | 34.0% | 65.5% | 88.3% |
| GPT-4o-mini | 74.6% | 52.5% | 14.3% | 34.0% | 55.0% | 74.4% |
| GPT-4o | **81.0%** | 63.8% | **50.4%** | **50.0%** | **69.4%** | 91.7% |
| *SFT with Synthetic Data* | | | | | | |
| Llama-2-7B-Chat | 40.6% | 11.3% | 15.8% | 32.0% | 30.6% | 93.7% |
| Llama-3-8B-Instruct | **63.5%** | **32.5%** | **33.0%** | **44.0%** | **51.1%** | **96.3%** |

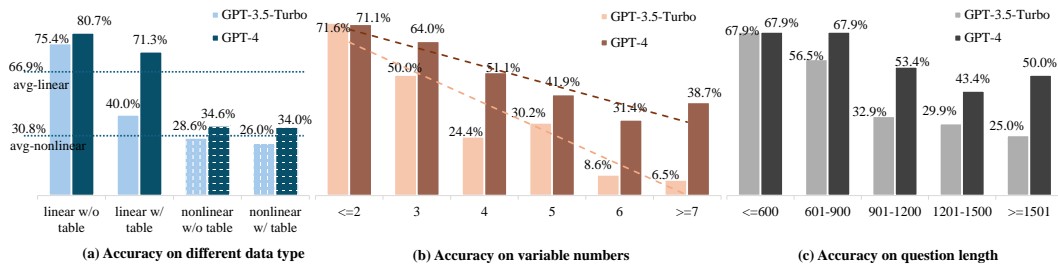

Figure 6: Performance analysis of GPT-4 and GPT-3.5-Turbo.

## 5.5 ABLATION STUDY ON **RESOCRATIC**

In Table 3, we present the performance outcomes of the models with and without the application of filters, specifically, the rule filter and similarity filter, which are illustrated in Figure 3(b). Additionally, we compare the performance when incorporating step questions versus when they are excluded. We conduct an ablation study on Llama-2-7B-Chat and Llama-3-8B-Instruct. The experimental results show the validity of our filters, this conclusion is similar to RFT (Yuan et al., 2023) that redundant data can negatively affect language models. Moreover, the step questions generated in **ReSocratic** can bring improvement to the model's solving ability.

Table 3: Ablation study on synthetic data.

| Model | SFT Data | Linear | | Nonlinear | | All |
|---|---|---|---|---|---|---|
| | | w/o Table | w/ Table | w/o Table | w/ Table | |
| Llama-2-7B-Chat | `ReSocratic` w/o step questions | 38.3% | 10.0% | 15.0% | 32.0% | 28.9% |
| | `ReSocratic` w/o filters | 40.1% | 11.3% | 14.3% | 30.0% | 29.6% |
| | RESOCRATIC-29K | **40.6%** | **11.3%** | **15.8%** | **32.0%** | **30.6%** |
| Llama-3-8B-Instruct | `ReSocratic` w/o step questions | 62.9% | 32.5% | 31.6% | 42.0% | 50.2% |
| | `ReSocratic` w/o filters | 62.3% | 31.3% | 32.3% | 36.0% | 49.4% |
| | RESOCRATIC-29K | **63.5%** | **32.5%** | **33.0%** | **44.0%** | **51.1%** |

## 5.6 COMPARISON BETWEEN REVERSE SYNTHESIS AND FORWARD SYNTHESIS

Furthermore, we compared the forward data synthesis approach with the reverse data synthesis approach (our **ReSocratic** method). WizardLM(Xu et al., 2023) is a typical forward data synthesis method, which first prompts the language model to generate questions similar to the seed data and then answer them. Using the same seed data, we sample 1000 responses from DeepSeek-v2 with wizardLM and **ReSocratic** respectively, and then fine-tune Llama-2-7B-Chat. The experimental results are shown in Table 4. The experimental results show that our **ReSocratic** synthesis method is superior to the forward synthesis method. Moreover, we sample 30 pieces of data generated by **ReSocratic** and WizardLM respectively, and manually identify the data accuracy, which also shows that the data generated by **ReSocratic** is more accurate.

Table 4: Comparison between **ReSocratic** and other forward methods (Evol-Instruct and Self-Instruct).

| Model | SFT Data | | Linear | | Nonlinear | | All |
|---|---|---|---|---|---|---|---|
| | Method | Data Acc | w/o Table | w/ Table | w/o Table | w/ Table | |
| Llama-2-7B-Chat | Self-Instruct (1k responses) | 80.0% | 16.1% | 5.0% | 3.0% | 4.0% | 10.7% |
| | Evol-Instruct (1k responses) | 76.7% | 15.5% | **7.5%** | 3.8% | 6.0% | 11.1% |
| | ReSocratic (1k responses) | **86.7%** | **21.6%** | 6.3% | **5.3%** | **6.0%** | **14.4%** |

## 6 CONCLUSION

In this paper, we propose the OPTIBENCH benchmark, which includes various types of data, to evaluate the ability of language models to solve mathematical optimization problems end-to-end. Furthermore, in order to alleviate the issue of data sparsity and mitigate the performance gap between large models and small open-source models, we introduce the **ReSocratic** method, a reverse data synthesis approach. The experimental results show that our **ReSocratic** method outperforms the forward data synthesis method. After fine-tuning with our synthetic data, RESOCRATIC-29K, the performance of Llama-2-7B-Chat and Llama-3-8B-Instruct has been significantly improved, demonstrating the effectiveness of our synthesis method. In the future, we plan to extend **ReSocratic** to other complex reasoning tasks such as math word problem-solving and evaluate more large language models on our proposed OPTIBENCH benchmark.

## ACKNOWLEDGMENTS

Jing Tang's work is partially supported by National Key R&D Program of China under Grant No. 2023YFF0725100 and No. 2024YFA1012701, by the National Natural Science Foundation of China (NSFC) under Grant No. 62402410 and No. U22B2060, by Guangdong Provincial Project (No. 2023QN10X025), by Guangdong Basic and Applied Basic Research Foundation under Grant No. 2023A1515110131, by Guangzhou Municipal Science and Technology Bureau under Grant No. 2023A03J0667 and No. 2024A04J4454, by Guangzhou Municipal Education Bureau (No. 2024312263), and by Guangzhou Municipality Big Data Intelligence Key Lab (No. 2023A03J0012), Guangzhou Industrial Information and Intelligent Key Laboratory Project (No. 2024A03J0628) and Guangzhou Municipal Key Laboratory of Financial Technology Cutting-Edge Research (No. 2024A03J0630).

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

APPENDIX

## A   MORE COMPARISONS WITH OTHER BENCHMARKS

**NL4OPT is too easy for current LLMs**. As we have mentioned in our paper, the NL4OPT benchmark is not an End-to-End evaluation benchmark. It conducts a two-stage process to evaluate the ability to construct mathematical models. In this work, we transform NL4OPT into our OPTIBENCH data format and contribute **NL4OPT-E**. We then evaluate a few LLMs, the results are shown in the following Table 5. All the experiments are conducted under the zero-shot setting. It is observed that LLMs solve NL4OPT almost completely even using the simplest prompt, but our OPTIBENCH still poses a strong challenge.

Table 5: Evaluation results comparison of NL4OPT-E and OPTIBENCH.

| Models | NL4OPT-E | OPTIBENCH |
|---|---|---|
| Deepseek-V2 | 89.3% | 34.4% |
| GPT-3.5-Turbo | 83.0% | 49.1% |
| GPT-4 | 93.1% | 62.8% |

**'Implicit' and 'Explicit' data**. Problems studied by Chain-of-Experts (Xiao et al., 2023) and OptiMUS (AhmadiTeshnizi et al., 2024) are orthogonal to ours. These related works (Xiao et al., 2023; AhmadiTeshnizi et al., 2024) examine the abstract modeling capabilities of LLMs for optimization problems. Specifically, the problems they focus on do not include explicit numerical values, primarily investigating the abstract modeling capabilities of LLMs in domain-specific scenarios. We denote this form of the problem as 'Implicit'. In contrast, the problems we study include explicit numbers, and researching the concrete problem-solving capabilities of LLMs. We denote this form of the problem as 'Explicit'. The form of the problems we study is closer to practical applications. We present an 'Implicit' sample and an 'Explicit' sample in Figure 7.

| **Implicit Sample** | **Explicit Sample** |
|---|---|
| A fishery wants to transport their catch. They can either use local sled dogs or trucks. Local sled dogs and trucks can take different amount of fish per trip. Also, the cost per trip for sled dogs and truck is also differs. You should note that the budget has an upper limit and the number of sled dog trips must be less than the number of truck trips. Formulate an LP to maximize the number of fish that can be transported. | A fishery wants to transport their catch. They can either use local sled dogs or trucks. Local sled dogs can take 100 fish per trip while trucks can take 300 fish per trip. The cost per trip for sled dogs is $50 while the cost per trip for a truck is $100. The budget is at most $1000 and the number of sled dog trips must be less than the number of truck trips. Formulate an LP to maximize the number of fish that can be transported. |

Figure 7: Examples of 'Implicit' and 'Explicit' data. The 'Implicit' sample is constructed by Xiao et al. (2023) according to the original 'Explicit' example of NL4OPT.

In this work, we mainly focus on the 'Explicit' problems. We consider such a form to be more related to real work scenarios, such as the query question containing a numerical table. Therefore, Chain-of-Experts (Xiao et al., 2023) and OptiMUS (AhmadiTeshnizi et al., 2024) are orthogonal to our work.

# B  ADDITIONAL EXPERIMENTAL RESULTS

## B.1  FINE-GRAINED ERROR ANALYSIS

We provide the statistical results on the Code Pass rate under various data categories. The results of Code Pass rate are shown as follows ('Code Pass' refers to the success rate of code execution).

Table 6: Code Pass rate under various data categories.

| Models | Prompt | Linear w/o Table | Linear w/ Table | Nonlinear w/o Table | Nonlinear w/ Table | Overall Pass |
|---|---|---|---|---|---|---|
| Mistral-7B-Instruct-v0.3 | zero-shot | 5.8 | 6.3 | 9.0 | 0.1 | 6.9 |
| Qwen2-7b-Instruct | zero-shot | 21.3 | 0.1 | 21.8 | 12.0 | 19.2 |
| gpt-4o | zero-shot | 93.6 | 93.8 | 83.5 | 78.0 | 90.1 |
| Mistral-7B-Instruct-v0.3 | few-shot | 87.7 | 85.0 | 75.9 | 84.0 | 83.8 |
| Qwen2-7b-Instruct | few-shot | 92.4 | 85.0 | 78.9 | 82.0 | 87.6 |
| gpt-4o | few-shot | 94.2 | 80.0 | 85.7 | 86.0 | 91.7 |

From the above statistical results, we can observe that the Code Pass rate for nonlinear problems is lower than that for linear problems, indicating that solving nonlinear problems poses certain challenges in terms of coding. Furthermore, we provide 2 case studies to show the errors in code generation by language models, as shown in the following.

## Error Case 1:

```python
import pyscipopt

# Create a new model
model = pyscipopt.Model()

# Define variables
acai_smoothies = model.addVar("acai_smoothies", vtype="I", lb=0)
banana_smoothies = model.addVar("banana_smoothies", vtype="I", lb=0)

# Define objective function
# Minimize the total amount of water used
obj = model.addVar('obj')
model.setObjective(obj, "minimize")
model.addCons(obj == 3 * acai_smoothies + 4 * banana_smoothies)

# Add constraints
# Constraint for acai berries
model.addCons(7 * acai_smoothies <= 3500)

# Constraint for banana chocolate
model.addCons(6 * banana_smoothies <= 3200)

# Constraint for more banana smoothies than acai smoothies
model.addCons(banana_smoothies > acai_smoothies)

# Constraint for at least 35% of smoothies being acai berry smoothies
model.addCons(acai_smoothies >= 0.35 * (acai_smoothies + banana_smoothies))

# Solve the problem
model.optimize()
```

## Error Analysis 1: PySCIPOpt does not support the symbols $<, >$; it only supports $\leq, \geq$.

## Error Case 2:

```python
import math
import pyscipopt

# Create a new model
model = pyscipopt.Model()

# Define variables for the rescue team's location (x, y)
x = model.addVar('x', lb=0)  # x must be non-negative
y = model.addVar('y')

# Define the man's location
```

```
man_x = 3/4
man_y = 0

# Define the objective function: minimize the distance to the man
distance = model.addVar('distance')
model.setObjective(distance, "minimize")  # We want to minimize the distance

# Add the distance constraint
model.addCons(distance == math.sqrt((x - man_x)**2 + (y - man_y)**2))

# Add the constraint for the swamp: y >= x^2
model.addCons(y >= x**2)

# Solve the problem
model.optimize()
```

## Error Analysis 2: In PySCIPOpt, you cannot use math.sqrt() directly.

Moreover, we measure the *accuracy of the mathematical formulations* using the LLM-as-a-judge approach and demonstrate the *accuracy of transferring the mathematical formulations to code*. We consider this analysis can provide a more comprehensive understanding of the models' performance. In this experiment, we use deepseek-v2.5 to judge the results.

Table 7: Accuracy of the Mathematical Formulations.

| Models | Prompt | Linear w/o Table | Linear w/ Table | Nonlinear w/o Table | Nonlinear w/ Table | Overall Pass |
|---|---|---|---|---|---|---|
| Mistral-7B-Instruct-v0.3 | zero-shot | 56.4 | 25.0 | 23.3 | 24.0 | 42.3 |
| Mistral-7B-Instruct-v0.3 | few-shot | 70.7 | 38.8 | 32.3 | 46.0 | 56.0 |
| Qwen2-7b-Instruct | zero-shot | 71.1 | 40.0 | 39.1 | 26.0 | 56.2 |
| Qwen2-7b-Instruct | few-shot | 78.7 | 43.8 | 41.4 | 46.0 | 63.1 |
| deepseek-v2.5 | zero-shot | 91.8 | 77.5 | 63.2 | 56.0 | 80.7 |
| deepseek-v2.5 | few-shot | 93.6 | 81.3 | 70.7 | 68.0 | 84.8 |
| gpt-4o-mini | zero-shot | 89.5 | 72.5 | 69.2 | 60.0 | 80.3 |
| gpt-4o-mini | few-shot | 90.1 | 67.5 | 72.9 | 70.0 | 81.7 |
| gpt-4o | zero-shot | 90.6 | 82.5 | 80.5 | 74.0 | 85.9 |
| gpt-4o | few-shot | 92.9 | 73.8 | 82.0 | 70.0 | 86.1 |

As shown in the table, small open-source models like Mistral-7B-Instruct-v0.3 and Qwen2-7b-Instruct tend to make more mistakes in mathematical formulations than large models like deepseek-v2.5 and gpt-4o. Moreover, the few-shot setting can significantly decrease the errors of mathematical formulations for all models than the zero-shot setting. These phenomena are also observed in benchmarks like GSM8K and MATH, leading us to conclude that they stem from the model's capacity for mathematical reasoning.

Table 8: Accuracy of Transferring the Mathematical Formulations to Code.

| Models | Prompt | Linear w/o Table | Linear w/ Table | Nonlinear w/o Table | Nonlinear w/ Table | Overall Pass |
|---|---|---|---|---|---|---|
| Mistral-7B-Instruct-v0.3 | zero-shot | 1.0 | 0.0 | 0.0 | 0.0 | 0.7 |
| Mistral-7B-Instruct-v0.3 | few-shot | 50.8 | 61.3 | 41.9 | 39.1 | 49.9 |
| Qwen2-7b-Instruct | zero-shot | 4.9 | 0.0 | 7.7 | 0.0 | 4.7 |
| Qwen2-7b-Instruct | few-shot | 83.3 | 62.9 | 45.5 | 30.4 | 72.8 |
| deepseek-v2.5 | zero-shot | 85.4 | 87.1 | 52.4 | 42.9 | 77.5 |
| deepseek-v2.5 | few-shot | 85.0 | 87.7 | 57.4 | 70.6 | 79.3 |
| gpt-4o-mini | zero-shot | 85.0 | 67.2 | 51.1 | 56.7 | 74.4 |
| gpt-4o-mini | few-shot | 82.8 | 77.8 | 19.6 | 48.6 | 67.4 |
| gpt-4o | zero-shot | 86.1 | 78.8 | 56.0 | 57.0 | 76.9 |
| gpt-4o | few-shot | 87.1 | 86.4 | 61.5 | 71.4 | 80.6 |

For models gpt-4o, gpt-4o-mini, and deepseek-v2.5, their errors in mathematical formulations are relatively few. gpt-4o achieves the sota performance on both two processes. Moreover, as can be seen from Table 8, the code transfer accuracy of Mistral-7B-Instruct-v0.3 and Qwen2-7b-Instruct is significantly lower in the zero-shot setting compared to the few-shot setting, which echoes our

previous observation (*models with weaker coding abilities are unable to correctly write code relying on their capabilities in a zero-shot setting but can solve problems correctly when provided with few-shot examples*).

In addition, we have conducted human-LLM consistency statistics. We randomly select 20 samples and ask 2 master's degree students in computer science to label them, comparing the labels to those of LLM-as-a-judge. The final results show that 19 out of 20 samples were consistent. This indicates the effectiveness of LLM-as-a-judge in fine-grained error analysis. The prompt of LLM-as-a-judge is shown in Appendix E.1.5.

## B.2 IMPACT OF DIFFERENT PROMPTING STRATEGIES

We investigate whether step-by-step reasoning then generates the code can help improve model performance. We construct a new prompt (first step reason then write code) named 'few-shot (first reason)'. The experimental results are shown in the following:

Table 9: Performance of Other Prompting Strategy.

| Models | Prompt | Overall Acc |
| --- | --- | --- |
| GPT-3.5-Turbo | few-shot (ori) | 56.4 |
| GPT-3.5-Turbo | few-shot (first reason) | 55.5 (-0.9) |
| GPT-4 | few-shot (ori) | 65.5 |
| GPT-4 | few-shot (first reason) | 63.8 (-1.7) |
| GPT-4o-mini | few-shot (ori) | 55.0 |
| GPT-4o-mini | few-shot (first reason) | 54.9 (-0.1) |
| GPT-4o | few-shot (ori) | 69.4 |
| GPT-4o | few-shot (first reason) | 67.1 (-2.3) |

'few-shot (ori)' is the original few-shot prompt in out paper, and 'few-shot (first reason)' is provided in the Appendix E.1.3. It can be seen from the results that "first understanding the reasoning and then writing code" does not significantly improve performance.

## B.3 PASS@K PERFORMANCE

We provide the Pass@k results as follows (we set the temperature as 0.7) as following. The performance of the model improves with the increase in the number of generation attempts.

Table 10: Pass@k Performance.

| Models | Pass@5 | Pass@10 | Pass@15 | Pass@20 | Pass@25 | Pass@30 |
| --- | --- | --- | --- | --- | --- | --- |
| Llama-3-8B-Instruct | 40.5% | 55.5% | 59.5% | 61.5% | 62.5% | 63.3% |
| Mistral-7B-Instruct-v0.3 | 47.1% | 57.0% | 61.3% | 63.8% | 64.7% | 66.1% |
| Qwen2-7b-Instruct | 59.5% | 63.5% | 65.8% | 67.3% | 68.3% | 68.6% |

## B.4 SOLVING EFFICIENCY OF THE GENERATED CODES

To fairly compare with code written by humans, for each tested LLM, we only compare the runtime of the LLM's code with that of human-written code on the samples where the LLM can provide correct answers. The statistical results are as follows (in seconds):

The aforementioned outcomes indicate that the efficiency of LLM written code is akin to that of human beings. The principal challenges for LLMs in solving optimization problems pertain to the accurate formulation of mathematical formulations and the generation of code that is devoid of errors.

Table 11: Comparison of Solving Efficiency

| Models | Prompt | LLM code runtime | Human code runtime |
|---|---|---|---|
| Mistral-7B-Instruct-v0.3 | zero-shot | 0.160 | 0.154 |
| Mistral-7B-Instruct-v0.3 | few-shot | 0.147 | 0.151 |
| Qwen2-7b-Instruct | zero-shot | 0.403 | 0.157 |
| Qwen2-7b-Instruct | few-shot | 0.212 | 0.143 |
| deepseek-v2.5 | zero-shot | 0.259 | 0.157 |
| deepseek-v2.5 | few-shot | 0.146 | 0.135 |
| gpt-4o-mini | zero-shot | 0.283 | 0.159 |
| gpt-4o-mini | few-shot | 0.114 | 0.145 |
| gpt-4o | zero-shot | 0.199 | 0.157 |
| gpt-4o | few-shot | 0.152 | 0.158 |

## B.5 POTENTIAL OF LLMS TO EXTEND PROBLEM COMPLEXITY

We conducte the following attempts: We provid a sample to DeepSeek-V2.5 and prompt the LLM to expand the number of variables and constraints of the sample. One of the expansion results is shown in the following:

Original Sample (6 variables, 3 constraints):

```
## Define Variables:
Gandhi Cloth Company is capable of manufacturing three types of clothing: shirts, shorts, and
    pants. The manufacture of each type of clothing requires Gandhi to rent the appropriate
    type of machinery. The company needs to determine the optimal number of each type of
    clothing to manufacture, and the number of each type of machinery to rent.
// {"number of shirts to manufacture": "Shirt", "range": "Shirt >= 0", "type": "integer"}
// {"number of shorts to manufacture": "Shorts", "range": "Shorts >= 0", "type": "integer"}
// {"number of pants to manufacture": "Pants", "range": "Pants >= 0", "type": "integer"}
// {"number of shirt machinery to rent": "Shirt_Machinery", "range": "Shirt_Machinery >= 0", "
    type": "integer"}
// {"number of shorts machinery to rent": "Shorts_Machinery", "range": "Shorts_Machinery >=
    0", "type": "integer"}
// {"number of pants machinery to rent": "Pants_Machinery", "range": "Pants_Machinery >= 0", "
    type": "integer"}

## Define Objective Function:
The revenue per shirt is $12, the revenue per shorts is $8, and the revenue per pants is $15.
The cost per shirt is $6, the cost per shorts is $4, and the cost per pants is $8.
The rental cost per shirt machinery per week is $200, the rental cost per shorts machinery per
    week is $150, and the rental cost per pants machinery per week is $100.
The company wants to maximize the weekly profit.
// Total_Revenue = 12*Shirt + 8*Shorts + 15*Pants
// Total_Cost = 6*Shirt + 4*Shorts + 8*Pants + 200*Shirt_Machinery + 150*Shorts_Machinery +
    100*Pants_Machinery
// Objective Function: Maximize: Total_Revenue - Total_Cost

## Generate Constraint-1:
The labor hours required per shirt is 3, the labor hours required per shorts is 2, and the
    labor hours required per pants is 6. Each week, 150 hours of labor are available.
// 3*Shirt + 2*Shorts + 6*Pants <= 150

## Generate Constraint-2:
The cloth required per shirt is 4 sq yd, the cloth required per shorts is 3 sq yd, and the
    cloth required per pants is 4 sq yd. Each week, 160 sq yd of cloth are available.
// 4*Shirt + 3*Shorts + 4*Pants <= 160

## Generate Constraint-3:
The company needs to rent at least one machinery for each type of clothing.
// Shirt_Machinery >= 1, Shorts_Machinery >= 1, Pants_Machinery >= 1
```

Expanded Samples (10 variables, 8 constraints):

```
### Define Variables:
Gandhi Cloth Company is capable of manufacturing five types of clothing: shirts, shorts, pants
    , jackets, and skirts. The manufacture of each type of clothing requires Gandhi to rent
    the appropriate type of machinery. The company needs to determine the optimal number of
    each type of clothing to manufacture, and the number of each type of machinery to rent.
// {"number of shirts to manufacture": "Shirt", "range": "Shirt >= 0", "type": "integer"}
```

```
// {"number of shorts to manufacture": "Shorts", "range": "Shorts >= 0", "type": "integer"}
// {"number of pants to manufacture": "Pants", "range": "Pants >= 0", "type": "integer"}
// {"number of jackets to manufacture": "Jackets", "range": "Jackets >= 0", "type": "integer"}
// {"number of skirts to manufacture": "Skirts", "range": "Skirts >= 0", "type": "integer"}
// {"number of shirt machinery to rent": "Shirt_Machinery", "range": "Shirt_Machinery >= 0", "
    type": "integer"}
// {"number of shorts machinery to rent": "Shorts_Machinery", "range": "Shorts_Machinery >=
    0", "type": "integer"}
// {"number of pants machinery to rent": "Pants_Machinery", "range": "Pants_Machinery >= 0", "
    type": "integer"}
// {"number of jacket machinery to rent": "Jacket_Machinery", "range": "Jacket_Machinery >=
    0", "type": "integer"}
// {"number of skirt machinery to rent": "Skirt_Machinery", "range": "Skirt_Machinery >= 0", "
    type": "integer"}

### Define Objective Function:
The revenue per shirt is $12, the revenue per shorts is $8, the revenue per pants is $15, the
    revenue per jacket is $20, and the revenue per skirt is $10.
The cost per shirt is $6, the cost per shorts is $4, the cost per pants is $8, the cost per
    jacket is $10, and the cost per skirt is $5.
The rental cost per shirt machinery per week is $200, the rental cost per shorts machinery per
     week is $150, the rental cost per pants machinery per week is $100, the rental cost per
    jacket machinery per week is $250, and the rental cost per skirt machinery per week is
    $120.
The company wants to maximize the weekly profit.
// Total_Revenue = 12*Shirt + 8*Shorts + 15*Pants + 20*Jackets + 10*Skirts
// Total_Cost = 6*Shirt + 4*Shorts + 8*Pants + 10*Jackets + 5*Skirts + 200*Shirt_Machinery +
    150*Shorts_Machinery + 100*Pants_Machinery + 250*Jacket_Machinery + 120*Skirt_Machinery
// Objective Function: Maximize: Total_Revenue - Total_Cost

### Generate Constraint-1:
The labor hours required per shirt is 3, the labor hours required per shorts is 2, the labor
    hours required per pants is 6, the labor hours required per jacket is 8, and the labor
    hours required per skirt is 4. Each week, 1500 hours of labor are available.
// 3*Shirt + 2*Shorts + 6*Pants + 8*Jackets + 4*Skirts <= 1500

### Generate Constraint-2:
The cloth required per shirt is 4 sq yd, the cloth required per shorts is 3 sq yd, the cloth
    required per pants is 4 sq yd, the cloth required per jacket is 5 sq yd, and the cloth
    required per skirt is 3 sq yd. Each week, 1600 sq yd of cloth are available.
// 4*Shirt + 3*Shorts + 4*Pants + 5*Jackets + 3*Skirts <= 1600

### Generate Constraint-3:
The company needs to rent at least one machinery for each type of clothing.
// Shirt_Machinery >= 1, Shorts_Machinery >= 1, Pants_Machinery >= 1, Jacket_Machinery >= 1,
    Skirt_Machinery >= 1

### Additional Constraints:
To ensure the problem remains feasible and interesting, we can add additional constraints
    based on the production capacity of each machinery type.

#### Constraint-4:
Each shirt machinery can produce up to 20 shirts per week.
// Shirt <= 20 * Shirt_Machinery

#### Constraint-5:
Each shorts machinery can produce up to 30 shorts per week.
// Shorts <= 30 * Shorts_Machinery

#### Constraint-6:
Each pants machinery can produce up to 15 pants per week.
// Pants <= 15 * Pants_Machinery

#### Constraint-7:
Each jacket machinery can produce up to 10 jackets per week.
// Jackets <= 10 * Jacket_Machinery

#### Constraint-8:
Each skirt machinery can produce up to 25 skirts per week.
// Skirts <= 25 * Skirt_Machinery
```

Furthermore, we had 3 master's degree students in computer science to examine 10 expanded samples. Specifically, the following points were checked:

Variables: Does the expanded variable relate to the optimization objective? Constraints: Are the expanded constraints reasonable? Formulation: Whether the formulation of the expanded mathematical model is correct. The verification results indicate that 9 out of the 10 expanded samples are correct.

We consider this to be the foundation for the model to self-improve its complex problem-solving abilities. In fact, this is also one of the future work plans we have.

## C  MORE DETAIL OF **ReSOCRATIC**

We have already shown the process of our synthesis method in our paper. This section adds more detail to our **ReSocratic**.

### C.1  SEED DEMONSTRATIONS

We collect 27 elaborate formatted demonstrations (13 linear scenarios and 14 nonlinear scenarios) in the seed pool. An example is shown in the following bellow.

```
## Define Variables:
Chip Green is the head groundskeeper at Birdie Valley Golf Club. There are four fertilizers (
    Fertilizer 1-4) available in the market, Chip would like to mix them together to obtain a
     mixture. Chip needs to determine the optimal proportion of each fertilizer in the
    mixture.
// {"proportion of Fertilizer 1 in the compost": "x1", "range": "0 <= x1 <= 1", "type": "
    continuous"}
// {"proportion of Fertilizer 2 in the compost": "x2", "range": "0 <= x2 <= 1", "type": "
    continuous"}
// {"proportion of Fertilizer 3 in the compost": "x3", "range": "0 <= x3 <= 1", "type": "
    continuous"}
// {"proportion of Fertilizer 4 in the compost": "x4", "range": "0 <= x4 <= 1", "type": "
    continuous"}
// The sum of the proportions should be 1: x1 + x2 + x3 + x4 = 1

## Define Objective Function:
The price of Fertilizer 1-4 is $21.75, $23.75, $22.00, and $19.50 per 100 pounds, respectively
    .
Chip wants to minimize the cost of the mixture per 100 pounds.
// Minimize: 21.75*x1 + 23.75*x2 + 22.00*x3 + 19.50*x4

## Generate Constraint-1:
The Nitrogen percentage of Fertilizer 1-4 is 10%, 8%, 12%, and 10%;
the Phosphorus percentage of Fertilizer 1-4 is 8%, 11%, 7%, and 10%;
the Potash percentage of Fertilizer 1-4 is 12, 15%, 12%, and 10%.
Chip knows that the best proportion of the chemical content should be 10-8-12 (10% nitrogen,
    8% phosphorus, and 12% potash), but no more than 0.5% above them. So the nitrogen level
    should be between 10% and 10.5%; the phosphorus level should be between 8% and 8.5%; the
    potash level should be between 12% and 12.5%.
// 10 <= 10*x1 + 8*x2 + 12*x3 + 10*x4 <= 10.5
// 8 <= 8*x1 + 11*x2 + 7*x3 + 10*x4 <= 8.5
// 12 <= 12*x1 + 15*x2 + 12*x3 + 10*x4 <= 12.5

## Generate Constraint-2:
The mixture should contain at least 20% Fertilizer 1.
// x1 >= 0.2
```

We show some statistical results of our formatted demonstration pool in Figure 8.

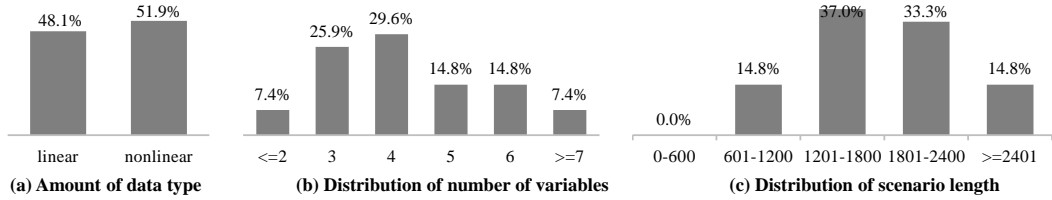

Figure 8: Statistical results of our demonstration pool.

We have shown the distribution of the formatted demonstrations of synthetic data in Figure 4. The data distribution of our synthetic dataset RESOCRATIC-29K is similar to the data distribution in our formatted demonstration pool.

## C.2 TABULAR DATA SYNTHESIZE

To facilitate the synthesis of different types of data (linear and nonlinear), we produce different prompts, which are shown in Section E.2.1 and Section E.2.2. In addition, in the back-translate stage, to get the question text with a table and the question text without a table, we construct two different back-translation prompts, as shown in Section E.2.3.

## C.3 BACK-TRANSLATION

We show an example of back-translation in the following:

Demonstration:

```
## Define Variables:
A cereal company makes nutritional cereal, kids' cereal, and sugary cereal. The company needs
    to determine the optimal number of boxes to produce for each type of cereal.
// {"number of nutritional cereal boxes": "x", "range": "x >= 0", "type": "integer"}
// {"number of kids' cereal boxes": "y", "range": "y >= 0", "type": "integer"}
// {"number of sugary cereal boxes": "z", "range": "z >= 0", "type": "integer"}

## Define Objective Function:
The revenue per box of nutritional cereal is $1, the revenue per kids' cereal is $1.50, and
    the revenue per sugary cereal is $2. How many of each should they make to maximize
    revenue?
// Maximize x + 1.5y + 2z

## Generate Constraint-1:
Each box of nutritional cereal requires 3 units of oat, each kids' cereal requires 1.5 units
    of oat, and each sugary cereal requires 2 units of oat. The company has available 500
    units of oat.
// 3x + 1.5y + 2z <= 500

## Generate Constraint-2:
Each box of nutritional cereal requires 1 unit of sugar, each kids' cereal requires 1.5 units
    of sugar, and each sugary cereal requires 4 units of sugar. The company has available 700
     units of sugar.
// x + 1.5y + 4z <= 700
```

Back-translated question:

```
A cereal company makes nutritional cereal, kids' cereal, and sugary cereal. The company needs
    to determine the optimal number of boxes to produce for each type of cereal. Each box of
    nutritional cereal requires 3 units of oat, each kids' cereal requires 1.5 units of oat,
    and each sugary cereal requires 2 units of oat. The company has available 500 units of
    oat. Each box of nutritional cereal requires 1 unit of sugar, each kids' cereal requires
    1.5 units of sugar, and each sugary cereal requires 4 units of sugar. The company has
    available 700 units of sugar. The revenue per box of nutritional cereal is $1, the
    revenue per kids' cereal is $1.50, and the revenue per sugary cereal is $2. How many of
    each should they make to maximize revenue?
```

As shown in this example, back-translation is almost equivalent to extracting the natural language part of the demonstration, which is a very simple task.

We asked 3 master's degree students in computer science to examine 20 back-translation samples to determine whether the generated questions in this step are correct. Specifically, the following points were checked:

- Whether the generated question text includes all the information from the demonstration.
- Whether the generated question is consistent with the question in the demonstration.

The test results show that all 20 samples are correct.

## C.4 TRAINING SAMPLE FORMAT

Specifically, we construct the training sample as follows:

```
[
    "system": "Please use python code with pyscipopt to solve the given optimization
        question."
    "user": "{Question}"
```

```
       "assistant":{Code}"
]
```

We replace "{*Question*}" and "{*Code*}" with the synthetic question and the verified code in our RESOCRATIC-29K to form the training sample.

# D  BENCHMARK AND DATASET

## D.1  MORE DETAILS OF DATA COLLECTION AND ANNOTATION

**Scale of Labeling Team**: Our labeling team consists of 6 professional annotators with master's degrees. Four of them are assigned to labeling tasks, while the other two annotators are responsible for correcting inconsistent labeling samples.

**Labeler Qualifications**:

- All team annotators have successfully passed the course of **Operations Research** and have a good theoretical foundation of mathematical modeling and optimization.
- All team annotators are proficient in Python programming language and the PySCIPOpt library.

**Source Data Details**:

- **Original data sources**: We assign annotators to collect questions from textbooks (Bertsimas & Tsitsiklis, 1997; Conforti et al., 2014; Wolsey, 2020), and a university's course assignments and examinations.
- **Data curation and annotation details**:
  1. First, the annotators are required to write the questions in markdown format. Simultaneously, record the question type (linear, nonlinear, with table, without table).
  2. Each question is annotated by 4 annotators, and each annotator performs the annotation independently. Specifically, the annotators are required to write the mathematical models in markdown format for those collected questions. If there are standard answers during the data collection process, we will instruct the annotators to directly record the mathematical model in the form of a formula in the markdown file. Otherwise, the annotators will be asked to write down the mathematical model themselves.
  3. After this step, the annotators were required to write code using PySCIPOpt to solve the problem and to record the values of the variables and the objective function.
  4. For each question, we compare the code execution results of the assigned 4 annotators. If the results are the same, the question is deemed as correctly labeled; otherwise, we will re-label and determine its final result.

## D.2  DATA FORMAT

**Data Format of OPTIBENCH.**  We store E-OPT data in the form of JSON files. A sample of our OPTIBENCH benchmark is shown below:

```
{
    "question": "A rectangular garden is to be constructed using a rock wall as one side of
        the garden and wire fencing for the other three sides. Given 100ft of wire fencing,
        determine the dimensions that would create a garden of maximum area. What is the
        maximum area?",
    "results": {
        "The length of the garden": "50.0",
        "The width of the garden": "25.0",
        "The maximum area of the garden": "1250.0"
    },
    "type": "nonlinear-notable",
    "index": 3
}
```

We construct samples in dictionary format, and all the data is stored as a list in a JSON file. Each sample has the following fields:

- **"question"**: The question text, presented in natural language, contains the background as well as the optimization objective and associated constraints. In order to solve the question, it is necessary to first find out the variables that can be optimized, then build a mathematical model, and then call a code solver to get the optimal numerical results of the variables and objective.

- **"results"**: This field is presented in the form of a dictionary, where the key is the natural language description of the variables and objectives, followed by their optimal values. During the annotation process, if the taggers cannot confirm that there is only one optimal solution to the problem, the results only contain the description of the optimization objective and its optimal value.

- **"type"**: This field records the type of the current sample, and there are four types: linear-table, linear-notable, non-linear-table, and nonlinear-notable.

- **"index"**: The index of the sample.

**Data Format of RESOCRATIC-29K.** We show a sample of our RESOCRATIC-29K in the following bellow.

```
{
    "question": "A logistics company operates four different routes for delivering packages.
        They need to determine the number of trucks to allocate to each route to optimize
        their operations. Each route has a different cost and revenue structure. On route 1,
        each truck incurs a cost of $100 per day and generates a revenue of $150 per day. On
        route 2, each truck incurs a cost of $120 per day and generates a revenue of $180 per
         day. On route 3, each truck incurs a cost of $140 per day and generates a revenue of
         $210 per day. On route 4, each truck incurs a cost of $160 per day and generates a
        revenue of $240 per day. The company aims to maximize the total daily profit across
        all routes. The company has a total of 50 trucks available. Please help the company
        to determine the optimal allocation of trucks to each route.",
    "code_solution": "import math\nimport pyscipopt\n\n# Create a new model\nmodel = pyscipopt
        .Model()\n\n# Define variables\nT1 = model.addVar(vtype=\"INTEGER\", name=\"T1\", lb=
        0) # number of trucks on route 1\nT2 = model.addVar(vtype=\"INTEGER\", name=\"T2\",
        lb=0) # number of trucks on route 2\nT3 = model.addVar(vtype=\"INTEGER\", name=\"T3\"
        , lb=0) # number of trucks on route 3\nT4 = model.addVar(vtype=\"INTEGER\", name=\"T4
        \", lb=0) # number of trucks on route 4\n\n# Define objective function\nProfit_route1
         = 150 * T1 - 100 * T1\nProfit_route2 = 180 * T2 - 120 * T2\nProfit_route3 = 210 * T3
         - 140 * T3\nProfit_route4 = 240 * T4 - 160 * T4\n# So, the objective function is:
        Maximize (Profit_route1 + Profit_route2 + Profit_route3 + Profit_route4)\nobj = model
        .addVar('obj')\nmodel.setObjective(obj, \"maximize\")\nmodel.addCons(obj ==
        Profit_route1 + Profit_route2 + Profit_route3 + Profit_route4)\n\n# Add constraints\n
        # The company has a total of 50 trucks available.\nmodel.addCons(T1 + T2 + T3 + T4 <=
         50)\n\n# Solve the problem\nmodel.optimize()\n\n# Print the optimal solution (value
        of the variables & the objective)\nprint('-'*10)\nif model.getStatus() == \"optimal\"
        :\n    print(\"Number of Trucks on Route 1: \", model.getVal(T1))\n    print(\"Number
         of Trucks on Route 2: \", model.getVal(T2))\n    print(\"Number of Trucks on Route 3
        : \", model.getVal(T3))\n    print(\"Number of Trucks on Route 4: \", model.getVal(T4
        ))\n    print(\"Maximized Total Daily Profit: \", model.getObjVal())\nelse:\n
        print(\"The problem could not be solved to optimality.\")\n"
}
```

We construct samples in dictionary format, and all the data is stored as a list in a JSON file. Each sample has the following fields:

- **"question"**: The question text, presented in natural language, contains the background as well as the optimization objective and associated constraints. In order to solve the question, it is necessary to first find out the variables that can be optimized, then build a mathematical model, and then call code solver to get the optimal numerical results of the variables and objective.

- **"code_solution"**: The corresponding python code to solve the question.

## E  ALL PROMPTS

We show all the prompts we used in this section.

## E.1 PROMPTS FOR EVALUATION OF OPTIBENCH

### E.1.1 ZERO-SHOT PROMPT

*"system":*

```
Please use python code to solve the given question.
```

*"user":*

```
[Code Template]:
'''python
import math
import pyscipopt

# Create a new model
model = pyscipopt.Model()

# Define variables
...

# Define objective function
## set objective as a variable (pyscipopt does not support non-linear objective)
obj = model.addVar('obj')
model.setObjective(obj, "...") # "maximize" or "minimize"
model.addCons(obj == ...) # obj function as a constraint

# Add constraints
...

# Solve the problem
model.optimize()

# Print the optimal solution (value of the variables & the objective)
print('-'*10)
if model.getStatus() == "optimal":
    ...
else:
    print("The problem could not be solved to optimality.")
'''

[Follow the code template to solve the given question, your code should be enclosed in '''
    python\n{}''']:
'''question
<... A testing question here ...>
'''
```

### E.1.2 FEW-SHOT PROMPT

*"system":*

```
Please follow the given examples and use python code to solve the given question.
```

*"user":*

```
[Example-1]:
'''question
A bakery specializes in producing two types of cakes: chocolate and vanilla. The bakery needs
    to decide how many of each type of cake to produce daily to maximize profit while
    considering the availability of ingredients and the minimum daily production requirement.
     The profit from each chocolate cake is $5, and from each vanilla cake is $4. The bakery
    aims to maximize its daily profit from cake sales. Each chocolate cake requires 2 eggs,
    and each vanilla cake requires 1 egg. The bakery has a daily supply of 100 eggs. Please
    help the bakery determine the optimal number of chocolate and vanilla cakes to produce
    daily.
'''

'''python
import math
import pyscipopt

# Create a new model
model = pyscipopt.Model()

# Define variables
```

```
## The number of each type of cake to produce daily
Choc = model.addVar(vtype="INTEGER", name="Choc", lb=0) # number of chocolate cakes
Van = model.addVar(vtype="INTEGER", name="Van", lb=0) # number of vanilla cakes

# Define objective function
## set objective as a variable
obj = model.addVar('obj')
model.setObjective(obj, "maximize")
model.addCons(obj == 5*Choc + 4*Van)

# Add constraints
## Each chocolate cake requires 2 eggs, and each vanilla cake requires 1 egg. The bakery has a
     daily supply of 100 eggs.
model.addCons(2*Choc + Van <= 100)

# Solve the problem
model.optimize()

# Print the optimal solution (value of the variables & the objective)
print('-'*10)
if model.getStatus() == "optimal":
    print("Number of chocolate cakes: ", model.getVal(Choc))
    print("Number of vanilla cakes: ", model.getVal(Van))
    print("Maximized Daily Profit: ", model.getObjVal())
else:
    print("The problem could not be solved to optimality.")
```

[Example-2]:
```question
A company produces three types of widgets: X, Y, and Z. The company needs to determine how
    many units of each widget to produce in next week.
For Widget X, the selling price is 10$, the material cost is 5$, and the production time is 2
    hours.
For Widget Y, the selling price is 15$, the material cost is 7$, and the production time is 3
    hours.
For Widget Z, the selling price is 20$, the material cost is 9$, and the production time is 4
    hours.
The company has $500 available for material costs next week. The company wants to produce at
    least 10 units of each widget next week. The company wants to spend at most 200 hours on
    production next week. The company has only one production line and can only produce one
    widget at a time. Please help the company to maximize the rate at which it earns profits
    (which is defined as the sum of the selling profit divided by the sum of the production
    times).
```

```python
import math
import pyscipopt

# Create a new model
model = pyscipopt.Model()

# Define variables
## The company wants to produce at least 10 units of each widget next week.
X = model.addVar(vtype="INTEGER", name="X", lb=10) # number of units of widget X
Y = model.addVar(vtype="INTEGER", name="Y", lb=10) # number of units of widget Y
Z = model.addVar(vtype="INTEGER", name="Z", lb=10) # number of units of widget Z

# Define objective function
## set objective as a variable (pyscipopt does not support non-linear objective)
obj = model.addVar('obj')
model.setObjective(obj, "maximize")
Profit_X = (10 - 5) * X
Profit_Y = (15 - 7) * Y
Profit_Z = (20 - 9) * Z
ProductionTime = 2 * X + 3 * Y + 4 * Z
## the objective function is: Maximize (Profit_X + Profit_Y + Profit_Z) / ProductionTime
## convert the division to multiplication
model.addCons(obj * ProductionTime == Profit_X + Profit_Y + Profit_Z)

# Add constraints
## The company has $500 available for material costs next week.
model.addCons(5 * X + 7 * Y + 9 * Z <= 500)
## The company wants to spend at most 200 hours on production next week.
model.addCons(2 * X + 3 * Y + 4 * Z <= 200)

# Solve the problem
model.optimize()
```

```
# Print the optimal solution (value of the variables & the objective)
print('-'*10)
if model.getStatus() == "optimal":
    print("Number of Widget X: ", model.getVal(X))
    print("Number of Widget Y: ", model.getVal(Y))
    print("Number of Widget Z: ", model.getVal(Z))
    print("Maximized Profit Rate: ", model.getObjVal())
else:
    print("The problem could not be solved to optimality.")
```

[Follow the examples to solve the given question]:
```question
<... A testing question here ...>
```

### E.1.3 FEW-SHOT (FIRST REASON) PROMPT

*"system":*

```
Please follow the given examples and use python code to solve the given question.
```

*"user":*

```
Given a question, you sould first establish the ```formulation``` step by step, then generate
    the ```python``` code to solve the problem using the 'pyscipopt' library.

[Example-1]:
Input:
```question
A bakery specializes in producing two types of cakes: chocolate and vanilla. The bakery needs
    to decide how many of each type of cake to produce daily to maximize profit while
    considering the availability of ingredients and the minimum daily production requirement.
     The profit from each chocolate cake is $5, and from each vanilla cake is $4. The bakery
    aims to maximize its daily profit from cake sales. Each chocolate cake requires 2 eggs,
    and each vanilla cake requires 1 egg. The bakery has a daily supply of 100 eggs. Please
    help the bakery determine the optimal number of chocolate and vanilla cakes to produce
    daily.
```

Your Output:
```formulation
## Define variables
* number of chocolate cakes
* number of vanilla cakes

## Define objective function
maximize the daily profit from cake sales: 5*Choc + 4*Van

## Add constraints
* Each chocolate cake requires 2 eggs, and each vanilla cake requires 1 egg. The bakery has a
    daily supply of 100 eggs. (2*Choc + Van <= 100)
```

```python
import math
import pyscipopt

model = pyscipopt.Model()

Choc = model.addVar(vtype="INTEGER", name="Choc", lb=0)
Van = model.addVar(vtype="INTEGER", name="Van", lb=0)

obj = model.addVar('obj')
model.setObjective(obj, "maximize")
model.addCons(obj == 5*Choc + 4*Van)

model.addCons(2*Choc + Van <= 100)

model.optimize()

print('-'*10)
if model.getStatus() == "optimal":
    print("Number of chocolate cakes: ", model.getVal(Choc))
    print("Number of vanilla cakes: ", model.getVal(Van))
    print("Maximized Daily Profit: ", model.getObjVal())
```

```
else:
    print("The problem could not be solved to optimality.")
```

[Example-2]:
Input:
```question
A company produces three types of widgets: X, Y, and Z. The company needs to determine how
    many units of each widget to produce in next week.
For Widget X, the selling price is 10$, the material cost is 5$, and the production time is 2
    hours.
For Widget Y, the selling price is 15$, the material cost is 7$, and the production time is 3
    hours.
For Widget Z, the selling price is 20$, the material cost is 9$, and the production time is 4
    hours.
The company has $500 available for material costs next week. The company wants to produce at
    least 10 units of each widget next week. The company wants to spend at most 200 hours on
    production next week. The company has only one production line and can only produce one
    widget at a time. Please help the company to maximize the rate at which it earns profits
    (which is defined as the sum of the selling profit divided by the sum of the production
    times).
```

Your Output:
```formulation
## Define variables
* number of units of widget X
* number of units of widget Y
* number of units of widget Z

## Define objective function
Maximize (Profit_X + Profit_Y + Profit_Z) / ProductionTime
where,
Profit_X = (10 - 5) * X
Profit_Y = (15 - 7) * Y
Profit_Z = (20 - 9) * Z
ProductionTime = 2 * X + 3 * Y + 4 * Z

## Add constraints
* The company has $500 available for material costs next week. (5 * X + 7 * Y + 9 * Z <= 500)
* The company wants to spend at most 200 hours on production next week. (2 * X + 3 * Y + 4 * Z
    <= 200)
* The company wants to produce at least 10 units of each widget next week. (X >= 10, Y >= 10,
    Z >= 10)
```

```python
import math
import pyscipopt

model = pyscipopt.Model()

X = model.addVar(vtype="INTEGER", name="X", lb=10) # number of units of widget X
Y = model.addVar(vtype="INTEGER", name="Y", lb=10) # number of units of widget Y
Z = model.addVar(vtype="INTEGER", name="Z", lb=10) # number of units of widget Z

obj = model.addVar('obj')
model.setObjective(obj, "maximize")
Profit_X = (10 - 5) * X
Profit_Y = (15 - 7) * Y
Profit_Z = (20 - 9) * Z
ProductionTime = 2 * X + 3 * Y + 4 * Z
model.addCons(obj * ProductionTime == Profit_X + Profit_Y + Profit_Z)

model.addCons(5 * X + 7 * Y + 9 * Z <= 500)
model.addCons(2 * X + 3 * Y + 4 * Z <= 200)

model.optimize()

print('-'*10)
if model.getStatus() == "optimal":
    print("Number of Widget X: ", model.getVal(X))
    print("Number of Widget Y: ", model.getVal(Y))
    print("Number of Widget Z: ", model.getVal(Z))
    print("Maximized Profit Rate: ", model.getObjVal())
else:
    print("The problem could not be solved to optimality.")
```

```
[Follow the examples to solve the given question]:
```
```

### E.1.4   RESULTS EXTRACTION PROMPT

```python
<... solution code generated by the LLM ...>
```

```code output
<... code execution result ...>
```

Accoding to the code output, please give your final answer for the following query. (The
    answer should be boxed in '\\boxed{}', and only in numerical form, and round it to 5
    decimal places, such as '\\boxed{27.00000}', '\\boxed{3.20000}', and '\\boxed{0.23334}').

<... query for the variables and objective ...>
```

### E.1.5   LLM AS A JUDGE PROMPT

*"system"*:

```
You are a math teacher. You are assessing whether a student's answer is correct. Specifically,
    you will be provided with a question, the ground truth mathematical formulation, and the
    student's code. What you need to do is determine whether the mathematical formulation in
    the student's code is correct based on the ground truth mathematical formulation. You
    need to first extract the mathematical formulation from the student's code and then
    compare it with the ground truth. You should express your final judgment as \boxed{
    Correct} or \boxed{Wrong}.
```

*"user"*:

```
## Question:
{question}

## Ground Truth Mathematical Formulation:
{ground_truth_formulation}

## Student Code:
{student_code}
```

## E.2   PROMPTS OF **ReSocratic**

### E.2.1   LINEAR DEMONSTRATION GENERATION

*"system"*:

```
Please follow the scenario examples to generate a [New Scenario] with a new background. The
    scenario should be a real-world linear optimization problem. Make sure that the
    mathematical logic in [New Scenario] is correct.
```

*"user"*:

```
[Scenario Format]:
## Define Variables:
natural language description.
// formal definition of variables (integer, real, binary, etc.) and their domains.

## Define Objective Function:
natural language description.
// formal definition of an objective function, maximize or minimize something. There can only
    be one objective function.

## Generate Constraint-1:
natural language description.
// formal definition of constraint-1

...

## Generate Constraint-n:
```

```
natural language description.
// formal definition of constraint-n

<... Sample 2 scenarios in the example pool ...>

[New Scenario]:
```

### E.2.2 NONLINEAR DEMONSTRATION GENERATION

*"system"*:

```
Please follow the scenario examples to generate a [New Scenario] with a new background. The
    scenario should be a real-world **nonlinear** optimization problem. Make sure that the
    mathematical logic in [New Scenario] is correct.
```

*"user"*:

```
[Scenario Format]:
## Define Variables:
natural language description.
// formal definition of variables (integer, real, binary, etc.) and their domains.

## Define Objective Function:
natural language description.
// formal definition of a **nonlinear** objective function, maximize or minimize something.
    There can only be one objective.

## Generate Constraint-1:
natural language description.
// formal definition of constraint-1

...

## Generate Constraint-n:
natural language description.
// formal definition of constraint-n

<... Sample 2 scenarios in the example pool ...>

[New Scenario]:
```

### E.2.3 QUESTION GENERATION

*"system"*:

```
You are a mathematical assistant. Now, you will be provided with an optimization scenario.
    Please follow the example to convert the given scenario to question.
```

**Generating questions without table.**

*"user"*:

```
[Task Description]:
You will be given a scenario that involves optimization problem. The scenario is organized
    into a few sections start with "##".
Each section contains a few lines of text that describe the scenario. The mathematical formal
    solution of the scenario is provided in the comments starting with "//".
Your job is to convert the scenario into a question without missing any information. The
    question should be clear and concise, and do not expose the mathematical formal solution
    of the scenario.

[Example of converting a Scenario to a Question]:
```scenario
## Define Variables:
A company produces five types of widgets: X, Y, Z, W, and V. The company needs to determine
    how many units of each widget to produce in next week.
// {"number of units of widget X": "X", "range": "X >= 0", "type": "integer"}
// {"number of units of widget Y": "Y", "range": "Y >= 0", "type": "integer"}
// {"number of units of widget Z": "Z", "range": "Z >= 0", "type": "integer"}
```

```
// {"number of units of widget W": "W", "range": "W >= 0", "type": "integer"}
// {"number of units of widget V": "V", "range": "V >= 0", "type": "integer"}

## Define Objective Function:
For Widget X, the selling price is $10, the material cost is $5, and the production time is 2
    hours.
For Widget Y, the selling price is $15, the material cost is $7, and the production time is 3
    hours.
For Widget Z, the selling price is $20, the material cost is $9, and the production time is 4
    hours.
For Widget W, the selling price is $25, the material cost is $11, and the production time is 5
     hours.
For Widget V, the selling price is $30, the material cost is $13, and the production time is 6
     hours.
The company has only one production line and can only produce one widget at a time. The
    company aims to maximize the rate at which it earns profits (which is defined as the sum
    of the selling profit divided by the sum of the production times).
// Selling profit of X: Profit_X = (10 - 5) * X
// Selling profit of Y: Profit_Y = (15 - 7) * Y
// Selling profit of Z: Profit_Z = (20 - 9) * Z
// Selling profit of W: Profit_W = (25 - 11) * W
// Selling profit of V: Profit_V = (30 - 13) * V
// So, the objective function is: Maximize (Profit_X + Profit_Y + Profit_Z + Profit_W +
    Profit_V) / (2 * X + 3 * Y + 4 * Z + 5 * W + 6 * V)

## Generate Constraint-1:
The company has $900 available for material costs next week.
// 5 * X + 7 * Y + 9 * Z + 11 * W + 13 * V <= 900

## Generate Constraint-2:
The company wants to produce at least 10 units of each widget next week.
// X >= 10; Y >= 10; Z >= 10; W >= 10; V >= 10

## Generate Constraint-3:
The company wants to spend at most 200 hours on production next week.
// 2 * X + 3 * Y + 4 * Z + 5 * W + 6 * V <= 200

## Generate Constraint-4:
The company wants to ensure that the total production of Widget W does not exceed the combined
     production of Widgets X, Y, and Z.
// W <= X + Y + Z
```

```question
A company produces five types of widgets: X, Y, Z, W, and V. The company needs to determine
    how many units of each widget to produce in next week.
For Widget X, the selling price is $10, the material cost is $5, and the production time is 2
    hours.
For Widget Y, the selling price is $15, the material cost is $7, and the production time is 3
    hours.
For Widget Z, the selling price is $20, the material cost is $9, and the production time is 4
    hours.
For Widget W, the selling price is $25, the material cost is $11, and the production time is 5
     hours.
For Widget V, the selling price is $30, the material cost is $13, and the production time is 6
     hours.
The company has $900 available for material costs next week. The company wants to produce at
    least 10 units of each widget next week. The company wants to spend at most 200 hours on
    production next week. The company wants to ensure that the total production of Widget W
    does not exceed the combined production of Widgets X, Y, and Z. The company has only one
    production line and can only produce one widget at a time.
Please help the company to maximize the rate at which it earns profits (which is defined as
    the sum of the selling profit divided by the sum of the production times).
```

[Follow the Example to Convert the following Scenario to a Question]:
```

**Generating questions with table.**

*"user"*:

```
[Task Description]:
You will be given a scenario that involves optimization problem. The scenario is organized
    into a few sections start with "##".
Each section contains a few lines of text that describe the scenario. The mathematical formal
    solution of the scenario is provided in the comments starting with "//".
Your job is to convert the scenario into a question without missing any information. The
    question should be clear and concise, and do not expose the mathematical formal solution
    of the scenario.
```

```
[Example of converting a Scenario to a Question with table]:
```scenario
## Define Variables:
A company produces five types of widgets: X, Y, Z, W, and V. The company needs to determine
    how many units of each widget to produce in next week.
// {"number of units of widget X": "X", "range": "X >= 0", "type": "integer"}
// {"number of units of widget Y": "Y", "range": "Y >= 0", "type": "integer"}
// {"number of units of widget Z": "Z", "range": "Z >= 0", "type": "integer"}
// {"number of units of widget W": "W", "range": "W >= 0", "type": "integer"}
// {"number of units of widget V": "V", "range": "V >= 0", "type": "integer"}

## Define Objective Function:
For Widget X, the selling price is $10, the material cost is $5, and the production time is 2
    hours.
For Widget Y, the selling price is $15, the material cost is $7, and the production time is 3
    hours.
For Widget Z, the selling price is $20, the material cost is $9, and the production time is 4
    hours.
For Widget W, the selling price is $25, the material cost is $11, and the production time is 5
     hours.
For Widget V, the selling price is $30, the material cost is $13, and the production time is 6
     hours.
The company has only one production line and can only produce one widget at a time. The
    company aims to maximize the rate at which it earns profits (which is defined as the sum
    of the selling profit divided by the sum of the production times).
// Selling profit of X: Profit_X = (10 - 5) * X
// Selling profit of Y: Profit_Y = (15 - 7) * Y
// Selling profit of Z: Profit_Z = (20 - 9) * Z
// Selling profit of W: Profit_W = (25 - 11) * W
// Selling profit of V: Profit_V = (30 - 13) * V
// So, the objective function is: Maximize (Profit_X + Profit_Y + Profit_Z + Profit_W +
    Profit_V) / (2 * X + 3 * Y + 4 * Z + 5 * W + 6 * V)

## Generate Constraint-1:
The company has $900 available for material costs next week.
// 5 * X + 7 * Y + 9 * Z + 11 * W + 13 * V <= 900

## Generate Constraint-2:
The company wants to produce at least 10 units of each widget next week.
// X >= 10; Y >= 10; Z >= 10; W >= 10; V >= 10

## Generate Constraint-3:
The company wants to spend at most 200 hours on production next week.
// 2 * X + 3 * Y + 4 * Z + 5 * W + 6 * V <= 200

## Generate Constraint-4:
The company wants to ensure that the total production of Widget W does not exceed the combined
     production of Widgets X, Y, and Z.
// W <= X + Y + Z
```

```question
A company produces five types of widgets: X, Y, Z, W, and V. The company needs to determine
    how many units of each widget to produce in next week. The selling price, material cost,
    and production time for each widget are given in the following Table.

| Widget | Selling Price | Material Cost | Production Time |
|--------|---------------|---------------|-----------------|
| X      | 10$           | 5$            | 2 hours         |
| Y      | 15$           | 7$            | 3 hours         |
| Z      | 20$           | 9$            | 4 hours         |
| W      | 25$           | 11$           | 5 hours         |
| V      | 30$           | 13$           | 6 hours         |

The company has $900 available for material costs next week. The company wants to produce at
    least 10 units of each widget next week. The company wants to spend at most 200 hours on
    production next week. The company wants to ensure that the total production of Widget W
    does not exceed the combined production of Widgets X, Y, and Z. The company has only one
    production line and can only produce one widget at a time.
Please help the company to maximize the rate at which it earns profits (which is defined as
    the sum of the selling profit divided by the sum of the production times).
```

[Follow the Example to Convert the following Scenario to a Question with table]:
```

### E.2.4 CODE GENERATION

*"system"*:

```
You are a mathematical assistant. Now, you will be provided with an optimization scenario with
    its corresponding question. Please follow the examples to solve the optimization
    scenario using python code with pyscipopt. (Tips: 1. Set objective as a variable to avoid
    non-linear objective. 2. To expedite computation, convert division to multiplication.)
```

*"user"*:

```
[Example-1]:
```scenario
## Define Variables:
Now we need to create a cylindrical metal jar with a metal shell.
// variables: {"radius of the cylindrical jar": "r", "height of the cylindrical jar": "h"},
    where r, h >= 0

## Define Objective Function:
The cost of the metal is $10 per square meter. Find the dimensions that will minimize the cost
    of the metal to manufacture the jar.
// The surface area of the cylindrical jar is the sum of the area of the two circular ends and
    the lateral surface area. The area of each circular end is \pi * r^2, and the lateral
    surface area is 2\pi*rh.
// So, the surface area of the cylindrical jar is 2\pi*r^2 + 2\pi*rh, and the cost of the
    metal is 10 * (2\pi*r^2 + 2\pi*rh).
// So, the objective function is: Minimize 10 * (2\pi*r^2 + 2\pi*rh)

## Generate Constraint-1:
The volume of the jar must be at least 1000 cubic centimeters.
// \pi*r^2h >= 1000
```

```python
import math
import pyscipopt

# Create a new model
model = pyscipopt.Model()

# Define variables
## The radius and height of the cylindrical jar
r = model.addVar(vtype="CONTINUOUS", name="r", lb=0, ub=100) # radius of the cylindrical jar
h = model.addVar(vtype="CONTINUOUS", name="h", lb=0, ub=100) # height of the cylindrical jar

# Define objective function
## set objective as a variable (pyscipopt does not support non-linear objective)
obj = model.addVar('obj')
model.setObjective(obj, "minimize")
## the objective function is: Minimize 10 * (2\pi*r^2 + 2\pi*rh)
model.addCons(obj == 10 * (2*math.pi*r**2 + 2*math.pi*r*h))

# Add constraints
## The volume of the jar must be at least 1000 cubic centimeters.
model.addCons(math.pi*r**2*h >= 1000)

# Solve the problem
model.optimize()

# Print the optimal solution (value of the variables & the objective)
print('-'*10)
if model.getStatus() == "optimal":
    print("Radius of the cylindrical jar: ", model.getVal(r))
    print("Height of the cylindrical jar: ", model.getVal(h))
    print("Minimized Cost: ", model.getObjVal())
else:
    print("The problem could not be solved to optimality.")
```

[Example-2]:
```scenario
## Define Variables:
You are designing a rectangular poster by cutting from a rectangular piece of paper.
// variables: {"width of the poster": "w", "height of the poster": "h"}, where w, h >= 0

## Define Objective Function:
The top and bottom margins are 2 inches, and the side margins are 1 inch. What dimensions of
    the poster should you use to minimize the area of paper used?
```

```
// The width of the used paper is w + 2*1, and the height of the used paper is h + 2*2.
// Therefore, the objective function is: Minimize (w + 2) * (h + 4)

## Generate Constraint-1:
The poster must have an area of 100 square inches.
// The area of the poster is given by the product of the width and the height, and it is given
     that the area is 100. Therefore, the constraint is w * h = 100
```

```python
import math
import pyscipopt

# Create a new model
model = pyscipopt.Model()

# Define variables
## The width and height of the poster
w = model.addVar(vtype="CONTINUOUS", name="w", lb=0, ub=100) # width of the poster
h = model.addVar(vtype="CONTINUOUS", name="h", lb=0, ub=100) # height of the poster

# Define objective function
## set objective as a variable (pyscipopt does not support non-linear objective)
obj = model.addVar('obj')
model.setObjective(obj, "minimize")
## the objective function is: Minimize (w + 2) * (h + 4)
model.addCons(obj == (w + 2) * (h + 4))

# Add constraints
## The poster must have an area of 100 square inches.
model.addCons(w * h == 100)

# Solve the problem
model.optimize()

# Print the optimal solution (value of the variables & the objective)
print('-'*10)
if model.getStatus() == "optimal":
    print("Width of the poster: ", model.getVal(w))
    print("Height of the poster: ", model.getVal(h))
    print("Minimized Area of Paper Used: ", model.getObjVal())
else:
    print("The problem could not be solved to optimality.")
```

[Convert the following Scenario to code]:
```scenario
<... Put your synthetic scenario here ...>
```
```

