# OpenReview forum: "OptiBench Meets ReSocratic: Measure and Improve LLMs for Optimization Modeling"
_ICLR.cc/2025/Conference — ICLR 2025 Poster_

### Official Review · Reviewer_PMkM · 2024-11-01

**Soundness:** 3
**Presentation:** 3
**Contribution:** 2
**Rating:** 6
**Confidence:** 3

**Summary:**

This paper proposes a study on the challenge of evaluating and improving large language models with respect to solving optimization problems. While the capabilities of large language models have recently improved with respect to mathematical and logical reasoning, the few existing benchmarks are limited to very simple tasks in linear programming and include little complexity representative of real-world applications. The authors present OPTIBENCH-a benchmark that attempts to evaluate the end-to-end ability of LLMs to solve optimization problems. It includes 605 diverse problems in the areas of linear and nonlinear programming, both tabular and nontabular; models are expected to generate code to execute and produce numerical answers.

Motivation: The authors mention that all existing optimization benchmarks are small-scaled and overly simplistic, where most of their problems are linear. Besides, open-sourced LLMs are substantially outperformed by more complex closed-source models like GPT-4. This demonstrates that a more challenging benchmark is needed along with efficient data augmentation methods to make smaller, open-source models perform better.
Main Contributions: OPTIBENCH Benchmark: The authors propose a new benchmark that includes various optimization problems, intended to test the LLM's capabilities in handling optimization tasks of varying complexity. The benchmark problems come in natural language formulation, which requires the generation of Python code that can be executed.
ReSocratic Data Synthesis Method: Based on this, the paper proposes ReSocratic-an approach for attempting to synthesize synthetic data in optimization. Unlike other methods, which first generate questions, the proposed ReSocratic works backward, constructing incremental step-by-step optimization demonstrations, and back-translates those into questions afterwards. This process aims to produce varied and consistent data. In this way, a dataset of questions, called RESOCRATIC-29K, is created.
Performance Improvement: This work presents data suggesting that fine-tuning open-source models over RESOCRATIC-29K might help narrow some of the gap with proprietary models. It reports accuracy improvements from 0.0% to 30.6% for Llama-2-7B-Chat and from 13.6% to 51.1% for Llama-3-8B-Instruct.
Evaluation and Analysis: The authors carried out extensive experiments in which the various models compared against each other in zero-shot, few-shot, and supervised fine-tuning settings. They mentioned nonlinear data and tabular data to be really challenging; increasing problem difficulty considerably raises the challenge, and their experiments indicate the ReSocratic method performs better than the straightforward forward synthesis techniques. Ablation studies further support the influence of key factors such as mechanisms for filtering and step-by-step synthesis.
The paper presents arguments for the potential utility of OPTIBENCH, and examines the effects of ReSocratic on open-source LLMs' optimization capabilities.

**Strengths:**

Originality: This work introduces the OPTIBENCH benchmark, a comprehensive tool designed to rigorously evaluate large language models (LLMs) on complex optimization problems, significantly advancing beyond existing benchmarks like MAMO and NLP4LP. Unlike prior efforts that primarily focus on simplistic, linear problems or abstract formulations, OPTIBENCH incorporates nonlinear and tabular data, simulating real-world scenarios more accurately. Additionally, the introduction of the ReSocratic data synthesis method is an inventive departure from traditional forward synthesis techniques, employing a reverse step-wise approach that yields a higher quality and diversity of synthetic optimization problems. While similar methods have been applied in other contexts, the adaptation and customization for optimization modeling distinguish this work meaningfully.

Quality: The paper is grounded in rigorous experimental design and analysis. The authors perform extensive evaluations using a variety of open-source and proprietary models, including detailed comparisons in zero-shot, few-shot, and fine-tuning settings. The results are compelling, showcasing substantial performance improvements, particularly when using ReSocratic-generated data, as demonstrated by a 30.6% accuracy improvement for Llama-2-7B-Chat and a 37.5% boost for Llama-3-8B-Instruct. The ablation studies and the analysis of data distribution and variable types further bolster the credibility of the experimental findings, highlighting the robustness and effectiveness of the proposed benchmark and synthesis method.

Clarity: The paper is well-structured and communicates its contributions clearly. The presentation of the OPTIBENCH benchmark, complete with illustrative examples of linear and nonlinear problems, aids in understanding its scope and novelty. The ReSocratic method is also carefully detailed, with step-by-step descriptions and visual aids that effectively convey the synthesis process. Moreover, the inclusion of a thorough comparison with existing benchmarks and methods provides a clear context for the advancements made.

Significance: The significance of this work lies in its practical applicability and potential impact on the field of optimization problem-solving with LLMs. By addressing the limitations of previous benchmarks and providing a more realistic and challenging testbed, OPTIBENCH has the potential to become a standard for evaluating and advancing LLMs in applied mathematics and operations research. Furthermore, the ReSocratic approach offers a scalable method to generate high-quality data, which could influence future research in data synthesis for complex reasoning tasks. The reported performance improvements also suggest practical benefits for enhancing open-source models, bridging the performance gap with proprietary models like GPT-4.

**Weaknesses:**

## Benchmark Contribution:
The present paper introduces OPTIBENCH for the evaluation of large language models in respect to solving optimization problems. However, such benchmarks already exist, for example MAMO, ComplexOR, and NLP4LP, which test LLMs concerning their ability to interpret natural language descriptions of optimization problems and create corresponding mathematical models. The extension in OPTIBENCH relates to nonlinear elements and tabular data. This feels like an incremental extension rather than a transformative one. The paper should better highlight the aspect of how OPTIBENCH goes significantly beyond the established benchmarks.
## Data Generation Process:
It proposes the data synthesis methodology, namely ReSocratic, and claims novelty in generating high-quality synthetic data. Earlier research has addressed reverse synthesis methodologies or back-translational techniques, especially for data augmentation and machine translation. Following are some related works which prove similarity:
"Data Augmentation for Code Generation Tasks" shows back-translation-style data augmentation that aims at improving code generation, just like ReSocratic. These techniques create synthetic variations from existing data. There are a number of similarities with this paper which raises the suspicion that the contribution of ReSocratic is more of an extension of the currently known methods and not genuinely new.
MathGenie: Synthetic Data Generation through Question Back-translation to Improve LLMs' Mathematics Reasoning, 2024: MathGenie synthesizes questions from mathematical solutions through a backward process, just like ReSocratic. Be that as it may, both methods are distinguished by their enhancing of reasoning through backward data synthesis, an issue that raises a question as to the novelty of ReSocratic in the given context.
"Data Augmentation for Code Translation with Comparable Corpora and Back-Translation" (2023): This work describes the generation of diverse code examples for translation tasks using back-translation, which aligns with ReSocratic's structured data synthesis approach. In fact, this would mean there is some correlation in the methodological procedure between the current work and existing strategies.
These related works demonstrate that such techniques as back-translation and reverse synthesis have already been successfully applied for code generation and mathematical reasoning. This paper could better explain how ReSocratic is different or improves these established methods. Moreover, discussing the possible biases of the reverse synthesis process would help to enhance the reliability and applicability of the generated data.
## Conclusion:
In general, this paper contains some interesting ideas; however, the contributions of both OPTIBENCH and the ReSocratic data generation method appear relatively modest with respect to prior work. It also shares similarities with approaches that have previously been proposed, indicating potential for better establishing the unique footprint and benefits introduced by the methods. An extended comparative study and some empirical evidence could be more enriching in showing points of strength for the proposed contributions.

**Questions:**

My major questions are around the data curation process.
# Data Labeling Process Questions

1. Scale of Labeling Team
- Total number of data labelers involved

2. Labeler Qualifications
- Required experience
- Necessary skills
- Background requirements

3. Source Data Details
- Data collection methods
- Criteria for benchmark curation
- Original data sources

---

> ### Author Response · Authors · 2024-11-23
> **Response to Reviewer PMkM (1/2)**
>
> We would like to thank the reviewer for the time and effort in reviewing our paper. We very much appreciate the insightful suggestions and your recognition of the originality, quality, clarity, and significance of our work.
>
> ### **1. Benchmark Contribution**
>
> Following your suggestions, we highlight the contribution of our benchmark and how it differs from other benchmarks as follows.
>
> * **Data Size**: Our benchmark contains **605** instances, which is significantly larger than these benchmarks: NL4OPT [1] with 289 instances, ComplexOR [2] with 37 instances, NLP4LP [3] with 67 instances.
>
> * **Data Type**: Our benchmark includes linear, nonlinear problems, and practical tabular format, which are not covered by existing benchmarks [1][2][3][4].
>
> * **‘Explicit’ Instance**: Problems studied by ComplexOR [2] and NLP4LP [3] are orthogonal to ours. Specifically, the problems they focus on do not include explicit numerical values, primarily investigating the abstract modeling capabilities of LLMs in domain-specific scenarios. We denote this form of the problem as ‘Implicit’. In contrast, the problems we study include explicit numbers and tables, and researching the concrete problem-solving capabilities of LLMs. We denote this form of the problem as ‘Explicit’. We consider the form of the problems we study to be closer to practical applications. We present an ’Implicit’ sample and an ’Explicit’ sample in the following.
>
> ### **2. Data Generation Process**
> Thank you for pointing out some related works [5][6][7]. I will elaborate on the differences between our work and these papers in the following:
>
> * *Code Translation Works* [5][7]: These two works both involve research on code translation, which is the process of translating one code language into another. Formally, given programming languages $PL_x$ and $PL_y$, to achieve the translation from $PL_x$ to $PL_y$, the core process of these two works involves using an existing model to back-translate $PL_y$ into $PL_x$, thereby constructing the synthetic data. Therefore, back-translation is the core module of these two works. However, our ReSocratic places more emphasis on step-by-step reverse construction of the chain-of-thought from scratch, with back-translation being one tiny step in our framework.
> * *MathGenie*[6]: This work is used for synthesizing mathematical reasoning data. It starts from the existing training data (the training set of GSM8k and MATH) and alters the ground truth solutions, then back-translates them into new problems. The work does not involve constructing synthetic data from scratch, and it requires a large amount of training data (7k of GSM and 7.5k of MATH) as a foundation.
>
> In summary, the aforementioned works [5][6][7] all start from existing data and construct synthetic data through back-translation. The significant difference between our work and the aforementioned is that we synthesize data **step-by-step** in a reverse manner **from scratch**.
>
> We cite the aforementioned works [5][6][7] and supplement these discussions in the Related Work of our revision.

---

> > ### Author Response · Authors · 2024-11-23
> > **An  ’Implicit’ Sample and An ’Explicit’ Sample**
> >
> > ### **Implicit Sample**
> > Question:
> > ```
> > A fishery wants to transport their catch. They can either use local sled dogs or trucks. Local sled dogs and trucks can take different amount of fish per trip. Also, the cost per trip for sled dogs and truck is also differs. You should note that the budget has an upper limit and the number of sled dog trips must be less than the number of truck trips. Formulate an LP to maximize the number of fish that can be transported.
> > ```
> >
> > ### **Explicit Sample (ours)**
> > Question:
> > ```
> > A company produces five types of widgets: X, Y, Z, W, and V. The company needs to determine how many units of each widget to produce in next week. The selling price, material cost, and production time for each widget are given in the following Table.
> >
> > | Widget | Selling Price | Material Cost | Production Time |
> > |--------|---------------|---------------|-----------------|
> > | X      | 10$           | 5$            | 2 hours         |
> > | Y      | 15$           | 7$            | 3 hours         |
> > | Z      | 20$           | 9$            | 4 hours         |
> > | W      | 25$           | 11$           | 5 hours         |
> > | V      | 30$           | 13$           | 6 hours         |
> >
> > The company has $900 available for material costs next week. The company wants to produce at least 10 units of each widget next week. The company wants to spend at most 200 hours on production next week. The company wants to ensure that the total production of Widget W does not exceed the combined production of Widgets X, Y, and Z. The company has only one production line and can only produce one widget at a time.
> > ```
> >
> > From the aforementioned examples, it can be observed that the implicit sample tests the model's abstract modeling capabilities without requiring the LLM to extract specific numerical values; whereas solving explicit samples requires the LLM to extract numerical information from the problem text and tables.
> > Therefore, we consider that the explicit form is more aligned with practical applications.

---

> ### Author Response · Authors · 2024-11-23
> **Response to Reviewer PMkM (2/2)**
>
> ### **3. Data Labeling Process**
> Thank you for pointing out the lack of details regarding Data Collection in our paper. We supplement the details as follows:
>
> **Scale of Labeling Team**: Our labeling team consists of 6 professional annotators with master's degrees. Four of them are assigned to labeling tasks, while the other 2 annotators are responsible for correcting inconsistent labeling samples.
>
> **Labeler Qualifications**:
> * All team annotators have successfully passed the course of **Operations Research** and have a good theoretical foundation of mathematical modeling and optimization.
> * All team annotators are proficient in Python programming language and the PySCIPOpt library.
>
> **Source Data Details**:
> * Original data sources: We assign annotators to collect questions from textbooks [8][9][10], assignments, and examinations from a university.
> * Data curation and annotation details:
>     1. First, the annotators are required to write the questions in markdown format. Simultaneously, record the question type (linear, nonlinear, with table, without table).
>     2. Each question is annotated by 4 annotators, and each annotator performs the annotation independently. Specifically, the annotators are required to write the mathematical models in markdown format for those collected questions. If there are standard answers during the data collection process, we will instruct the annotators to directly record the mathematical model in the form of a formula in the markdown file. Otherwise, the annotators will be asked to write down the mathematical model themselves.
>     3. After this step, the annotators were required to write code using PySCIPOpt to solve the problem and to record the values of the variables and the objective function.
>     4. For each question, we compare the code execution results of the assigned 4 annotators. If the results are the same, the question is deemed as correctly labeled; otherwise, we will re-label and determine its final result.
>
> We incorporate these contents into Appendix D.1 of our manuscript.

---

> ### Author Response · Authors · 2024-11-23
> **References**
>
> [1] Ramamonjison R, Yu T, Li R, et al. Nl4opt competition: Formulating optimization problems based on their natural language descriptions. NeurIPS 2022 Competition Track. PMLR, 2023: 189-203.
> [2] Xiao Z, Zhang D, Wu Y, et al. Chain-of-Experts: When LLMs Meet Complex Operations Research Problems. ICLR 2024.
> [3] AhmadiTeshnizi A, Gao W, Udell M. OptiMUS: Scalable Optimization Modeling with (MI) LP Solvers and Large Language Models. arXiv preprint arXiv:2402.10172, 2024.
> [4] Huang X, Shen Q, Hu Y, et al. Mamo: a Mathematical Modeling Benchmark with Solvers. arXiv preprint arXiv:2405.13144, 2024.
> [5] Chen P, Lampouras G. Exploring data augmentation for code generation tasks. arXiv preprint arXiv:2302.03499, 2023.
> [6] Lu Z, Zhou A, Ren H, et al. Mathgenie: Generating synthetic data with question back-translation for enhancing mathematical reasoning of llms. arXiv preprint arXiv:2402.16352, 2024.
> [7] Xie Y, Naik A, Fried D, et al. Data Augmentation for Code Translation with Comparable Corpora and Multiple References. arXiv preprint arXiv:2311.00317, 2023.
> [8] Dimitris Bertsimas and John N Tsitsiklis. Introduction to linear optimization, volume 6. Athena Scientific Belmont, MA, 1997.
> [9] Michele C, Gerard C, Giacomo Z, et al. Integer programming models. Springer, 2014.
> [10] Laurence A Wolsey. Integer programming. John Wiley & Sons, 2020.

---

### Official Review · Reviewer_GR55 · 2024-11-01

**Soundness:** 3
**Presentation:** 3
**Contribution:** 3
**Rating:** 6
**Confidence:** 4

**Summary:**

This paper proposes a benchmark for end-to-end optimization problem-solving and a data synthesis method. The benchmark contains various optimization problems, including linear and non-linear programming with or without tabular data. With the constructed benchmark, the fine-tuned LLMs exhibit a stronger performance.

**Strengths:**

- This paper proposes a benchmark containing various problems, including linear and non-linear programming with or without tabular data, which can better evaluate the ability of LLMs.
- The reverse data synthesis approach is novel and reasonable.
- The experimental results show that ReSocratic outperforms the forward data synthesis method, and the fine-tuning results are promising.

**Weaknesses:**

- The authors may want to generate instances with more constraints and variables, as few instances in the paper have more than 7 variables. Thus, this raises my concern about LLMs' ability to model problems with large instance sizes.
- Given that a single optimization problem can have multiple valid formulations, it would be beneficial for the authors to verify the accuracy and equivalence of these formulations with ground-truth ones.
- There are questions regarding the solving efficiency of the generated codes. It would be valuable to assess whether the code produced by LLMs can outperform human-designed formulations and codes.

**Questions:**

- Could you please provide some results on LLMs handling the large-size problem instances?
- Could you please verify the accuracy of the formulation given by the LLMs?
- Would it be possible to compare the solving efficiency of the codes produced by the LLMs?

---

> ### Author Response · Authors · 2024-11-23
> **Response to Reviewer GR55**
>
> Thank you for your time and effort in reviewing our paper. We very much appreciate your acknowledgement of our ***OptiBench*** and ***ReScoratic***. We hereby address the concerns below:
>
>
> ### **1. Formulation Accuracy**
>
> Thank you for pointing out that the accuracy of the formulation is important.
> We provide the statistical results on the Code Pass rate under various data categories to indirectly illustrate the effectiveness of the model formulation. The natural language form of the mathematical formulation cannot be analyzed directly by exact matching with ground truth labels. It requires manual annotation and is time-consuming.
>
> The results of Code Pass rate are shown as follows ('Code Pass' refers to the success rate of code execution):
>
> | Models | Prompt |  Linear w/o Table | Linear w/ Table | Nonlinear w/o Table | Nonlinear w/ Table | Overall Pass |
> | --- | --- | :---: | :---: | :---: | :---: | :---: |
> | ***zero-shot setting*** |
> | Mistral-7B-Instruct-v0.3 | zero-shot |5.8| 6.3| 9.0| 0.1| 6.9|
> | Qwen2-7b-Instruct | zero-shot |21.3| 0.1| 21.8| 12.0| 19.2 |
> | gpt-4o | zero-shot | 93.6 | 93.8 | 83.5 | 78.0 | 90.1 |
> | ***few-shot setting*** |
> | Mistral-7B-Instruct-v0.3 | few-shot |87.7| 85.0| 75.9| 84.0| 83.8|
> | Qwen2-7b-Instruct | few-shot | 92.4| 85.0| 78.9| 82.0| 87.6|
> | gpt-4o | few-shot | 94.2 | 80.0 |  85.7 | 86.0 | 91.7 |
>
> From the above statistical results, we can observe that the Code Pass rate for nonlinear problems is lower than that for linear problems, indicating that solving nonlinear problems poses certain challenges in terms of coding.
>
> We add the experimental results to Table 6 of our revision.
>
>
> ### **2. Solving Efficiency of the Generated Codes**
>
> We agree that solving efficiency is a point worth noting. To fairly compare with code written by humans, for each tested LLM, we only compare the runtime of the LLM's code with that of human-written code on the samples where the LLM can provide correct answers.
> The statistical results are as follows (in seconds):
>
> | Models | Prompt | LLM code runtime | Human code runtime |
> | --- | --- | :---: | :---: |
> | Mistral-7B-Instruct-v0.3 | zero-shot | 0.160 | 0.154 |
> | Mistral-7B-Instruct-v0.3 | few-shot | 0.147 | 0.151 |
> | Qwen2-7b-Instruct | zero-shot | 0.403 | 0.157 |
> | Qwen2-7b-Instruct | few-shot | 0.212 | 0.143 |
> | deepseek-v2.5 | zero-shot | 0.259 | 0.157 |
> | deepseek-v2.5 | few-shot | 0.146 | 0.135 |
> | gpt-4o-mini | zero-shot | 0.283 | 0.159 |
> | gpt-4o-mini | few-shot | 0.114 | 0.145 |
> | gpt-4o | zero-shot | 0.199 | 0.157 |
> | gpt-4o | few-shot | 0.152 | 0.158 |
>
> We add the experimental results to Table 11 of our revision.
> The aforementioned outcomes indicate that the efficiency of LLM written code is akin to that of human beings. The principal challenges for LLMs in solving optimization problems pertain to the accurate formulation of mathematical formulations and the generation of code that is devoid of errors.
>
> ### **3. Generate Large-Size Problem Instances with More Constraints and Variables**
>
> We agree that the large-size problem is crucial for the application of LLM in Operations Research. We conducte the following attempts: We provid a sample to DeepSeek-V2.5 and prompt the LLM to expand the number of variables and constraints of the sample. One of the expansion results is shown in the attached comment.
>
> Furthermore, we had 3 master's degree students in computer science to examine 10 expanded samples. Specifically, the following points were checked:
> * Variables: Does the expanded variable relate to the optimization objective?
> * Constraints: Are the expanded constraints reasonable?
> * Formulation: Whether the formulation of the expanded mathematical model is correct.
>
>
> The verification results indicate that 9 out of the 10 expanded samples are correct.
>
> We consider this to be the foundation for the model to self-improve its complex problem-solving abilities. In fact, this is also one of the future work plans we have.

---

> > ### Author Response · Authors · 2024-11-23
> > **Large-Size Problem Expansion Example (Original Sample)**
> >
> > **Original Sample (6 variables, 3 constraints)**:
> > ```
> > ## Define Variables:
> > Gandhi Cloth Company is capable of manufacturing three types of clothing: shirts, shorts, and pants. The manufacture of each type of clothing requires Gandhi to rent the appropriate type of machinery. The company needs to determine the optimal number of each type of clothing to manufacture, and the number of each type of machinery to rent.
> > // {"number of shirts to manufacture": "Shirt", "range": "Shirt >= 0", "type": "integer"}
> > // {"number of shorts to manufacture": "Shorts", "range": "Shorts >= 0", "type": "integer"}
> > // {"number of pants to manufacture": "Pants", "range": "Pants >= 0", "type": "integer"}
> > // {"number of shirt machinery to rent": "Shirt_Machinery", "range": "Shirt_Machinery >= 0", "type": "integer"}
> > // {"number of shorts machinery to rent": "Shorts_Machinery", "range": "Shorts_Machinery >= 0", "type": "integer"}
> > // {"number of pants machinery to rent": "Pants_Machinery", "range": "Pants_Machinery >= 0", "type": "integer"}
> >
> > ## Define Objective Function:
> > The revenue per shirt is $12, the revenue per shorts is $8, and the revenue per pants is $15.
> > The cost per shirt is $6, the cost per shorts is $4, and the cost per pants is $8.
> > The rental cost per shirt machinery per week is $200, the rental cost per shorts machinery per week is $150, and the rental cost per pants machinery per week is $100.
> > The company wants to maximize the weekly profit.
> > // Total_Revenue = 12*Shirt + 8*Shorts + 15*Pants
> > // Total_Cost = 6*Shirt + 4*Shorts + 8*Pants + 200*Shirt_Machinery + 150*Shorts_Machinery + 100*Pants_Machinery
> > // Objective Function: Maximize: Total_Revenue - Total_Cost
> >
> > ## Generate Constraint-1:
> > The labor hours required per shirt is 3, the labor hours required per shorts is 2, and the labor hours required per pants is 6. Each week, 150 hours of labor are available.
> > // 3*Shirt + 2*Shorts + 6*Pants <= 150
> >
> > ## Generate Constraint-2:
> > The cloth required per shirt is 4 sq yd, the cloth required per shorts is 3 sq yd, and the cloth required per pants is 4 sq yd. Each week, 160 sq yd of cloth are available.
> > // 4*Shirt + 3*Shorts + 4*Pants <= 160
> >
> > ## Generate Constraint-3:
> > The company needs to rent at least one machinery for each type of clothing.
> > // Shirt_Machinery >= 1, Shorts_Machinery >= 1, Pants_Machinery >= 1
> > ```

---

> > ### Author Response · Authors · 2024-11-23
> > **Large-Size Problem Expansion Example (Expanded Sample)**
> >
> > **Expanded Samples (10 variables, 8 constraints)**:
> > ```
> > ### Define Variables:
> > Gandhi Cloth Company is capable of manufacturing five types of clothing: shirts, shorts, pants, jackets, and skirts. The manufacture of each type of clothing requires Gandhi to rent the appropriate type of machinery. The company needs to determine the optimal number of each type of clothing to manufacture, and the number of each type of machinery to rent.
> > // {"number of shirts to manufacture": "Shirt", "range": "Shirt >= 0", "type": "integer"}
> > // {"number of shorts to manufacture": "Shorts", "range": "Shorts >= 0", "type": "integer"}
> > // {"number of pants to manufacture": "Pants", "range": "Pants >= 0", "type": "integer"}
> > // {"number of jackets to manufacture": "Jackets", "range": "Jackets >= 0", "type": "integer"}
> > // {"number of skirts to manufacture": "Skirts", "range": "Skirts >= 0", "type": "integer"}
> > // {"number of shirt machinery to rent": "Shirt_Machinery", "range": "Shirt_Machinery >= 0", "type": "integer"}
> > // {"number of shorts machinery to rent": "Shorts_Machinery", "range": "Shorts_Machinery >= 0", "type": "integer"}
> > // {"number of pants machinery to rent": "Pants_Machinery", "range": "Pants_Machinery >= 0", "type": "integer"}
> > // {"number of jacket machinery to rent": "Jacket_Machinery", "range": "Jacket_Machinery >= 0", "type": "integer"}
> > // {"number of skirt machinery to rent": "Skirt_Machinery", "range": "Skirt_Machinery >= 0", "type": "integer"}
> >
> > ### Define Objective Function:
> > The revenue per shirt is $12, the revenue per shorts is $8, the revenue per pants is $15, the revenue per jacket is $20, and the revenue per skirt is $10.
> > The cost per shirt is $6, the cost per shorts is $4, the cost per pants is $8, the cost per jacket is $10, and the cost per skirt is $5.
> > The rental cost per shirt machinery per week is $200, the rental cost per shorts machinery per week is $150, the rental cost per pants machinery per week is $100, the rental cost per jacket machinery per week is $250, and the rental cost per skirt machinery per week is $120.
> > The company wants to maximize the weekly profit.
> > // Total_Revenue = 12*Shirt + 8*Shorts + 15*Pants + 20*Jackets + 10*Skirts
> > // Total_Cost = 6*Shirt + 4*Shorts + 8*Pants + 10*Jackets + 5*Skirts + 200*Shirt_Machinery + 150*Shorts_Machinery + 100*Pants_Machinery + 250*Jacket_Machinery + 120*Skirt_Machinery
> > // Objective Function: Maximize: Total_Revenue - Total_Cost
> >
> > ### Generate Constraint-1:
> > The labor hours required per shirt is 3, the labor hours required per shorts is 2, the labor hours required per pants is 6, the labor hours required per jacket is 8, and the labor hours required per skirt is 4. Each week, 1500 hours of labor are available.
> > // 3*Shirt + 2*Shorts + 6*Pants + 8*Jackets + 4*Skirts <= 1500
> >
> > ### Generate Constraint-2:
> > The cloth required per shirt is 4 sq yd, the cloth required per shorts is 3 sq yd, the cloth required per pants is 4 sq yd, the cloth required per jacket is 5 sq yd, and the cloth required per skirt is 3 sq yd. Each week, 1600 sq yd of cloth are available.
> > // 4*Shirt + 3*Shorts + 4*Pants + 5*Jackets + 3*Skirts <= 1600
> >
> > ### Generate Constraint-3:
> > The company needs to rent at least one machinery for each type of clothing.
> > // Shirt_Machinery >= 1, Shorts_Machinery >= 1, Pants_Machinery >= 1, Jacket_Machinery >= 1, Skirt_Machinery >= 1
> >
> > ### Additional Constraints:
> > To ensure the problem remains feasible and interesting, we can add additional constraints based on the production capacity of each machinery type.
> >
> > #### Constraint-4:
> > Each shirt machinery can produce up to 20 shirts per week.
> > // Shirt <= 20 * Shirt_Machinery
> >
> > #### Constraint-5:
> > Each shorts machinery can produce up to 30 shorts per week.
> > // Shorts <= 30 * Shorts_Machinery
> >
> > #### Constraint-6:
> > Each pants machinery can produce up to 15 pants per week.
> > // Pants <= 15 * Pants_Machinery
> >
> > #### Constraint-7:
> > Each jacket machinery can produce up to 10 jackets per week.
> > // Jackets <= 10 * Jacket_Machinery
> >
> > #### Constraint-8:
> > Each skirt machinery can produce up to 25 skirts per week.
> > // Skirts <= 25 * Skirt_Machinery
> > ```

---

> ### Author Response · Authors · 2024-11-26
> **Additional Response About Formulation Accuracy**
>
> Inspired by Reviewer hRyJ , we have been delving deeper into error analysis. Finally, through our relentless efforts, we are able to measure the *accuracy of the mathematical formulations* using the "**LLM as a judge**" approach. Moreover, we have conducted human-LLM consistency statistics to confirm our results. This analysis can provide a more comprehensive understanding of the models' performance.
>
> In our experiment, we use deepseek-v2.5 to judge the results. The experimental results are shown in the following additional tables.
>
> **Additional Table 1: Accuracy of the Mathematical Formulations.**
>
> | Models | Prompt |  Linear w/o Table | Linear w/ Table | Nonlinear w/o Table | Nonlinear w/ Table | Overall |
> | --- | --- | :---: | :---: | :---: | :---: |  :---:  |
> | Mistral-7B-Instruct-v0.3 | zero-shot | 56.4 | 25.0 | 23.3 | 24.0 | 42.3 |
> | Mistral-7B-Instruct-v0.3 | few-shot | 70.7 | 38.8 | 32.3 | 46.0 | 56.0 |
> | Qwen2-7b-Instruct | zero-shot | 71.1 | 40.0 | 39.1 | 26.0 | 56.2 |
> | Qwen2-7b-Instruct | few-shot  | 78.7 | 43.8 | 41.4 |46.0 | 63.1 |
> | deepseek-v2.5 | zero-shot | 91.8 | 77.5 | 63.2 | 56.0 | 80.7 |
> | deepseek-v2.5 | few-shot  | 93.6 | 81.3 | 70.7 | 68.0 | 84.8 |
> | gpt-4o-mini | zero-shot | 89.5 | 72.5 | 69.2 | 60.0 | 80.3 |
> | gpt-4o-mini | few-shot  | 90.1 | 67.5 | 72.9 | 70.0 | 81.7 |
> | gpt-4o | zero-shot | 90.6 | 82.5 | 80.5 | 74.0 | 85.9 |
> | gpt-4o | few-shot  | 92.9 | 73.8 | 82.0 | 70.0 | 86.1 |
>
>
> As shown in the table, small open-source models like Mistral-7B-Instruct-v0.3 and Qwen2-7b-Instruct tend to make more mistakes in mathematical formulations than large models like deepseek-v2.5 and gpt-4o. Moreover, the few-shot setting can significantly decrease the errors of mathematical formulations for all models than the zero-shot setting. These phenomena are also observed in benchmarks like GSM8K and MATH, leading us to conclude that they stem from the model's capacity for mathematical reasoning.
>
> We add these experimental results to Table 7 of our revision. The judge prompt is shown in Appendix E.1.5.
>
>
> **Human-LLM consistency Statistics**
>
> In addition, we have conducted human-LLM consistency statistics. We randomly select 20 samples and ask 2 master's degree students in computer science to label them, comparing the labels to those of "LLM as a judge". The final results show that 19 out of 20 samples were consistent. This indicates the effectiveness of "LLM as a judge" in fine-grained error analysis.

---

### Official Review · Reviewer_hRyJ · 2024-11-10

**Soundness:** 2
**Presentation:** 3
**Contribution:** 3
**Rating:** 8
**Confidence:** 3

**Summary:**

This paper proposes OPTIBENCH, an end-to-end benchmark for evaluating large language models (LLMs) on optimization modeling. OPTIBENCH covers various problem types, including linear and nonlinear programming with/without tabular data. To further alleviate data scarcity, the authors introduce ReSocratic, a `reverse` data synthesis method that generates the RESOCRATIC-29K dataset. The authors demonstrated that fine-tuning open-source LLMs (Llama2 7b and Llama3 8B) on RESOCRATIC-29K significantly improves their performance on OPTIBENCH. The paper contributes a comprehensive benchmark, a reverse data synthesis method, and a synthetic dataset to advance evaluations on optimization modeling with LLMs.

**Strengths:**

OPTIBENCH is a comprehensive benchmark that effectively evaluates the optimization problem-solving abilities. It inlcudes nonlinear programming problems, along with tabular data, reflecting realistic scenarios. It also makes the benchmark challenging enough.
By requiring LLMs to understand the problem, perform sound reasoning, and generate code to invoke a solver, OPTIBENCH provides a holistic assessment of LLMs' reasoning and coding skills.
This general evaluation approach is valuable for measuring the models' ability to apply their knowledge to practical scenarios and can guide future research on LLMs' capabilities in reasoning tasks. Moreover, the data annotation with strong human guidance looks reliable.

The benchmark evaluates the correctness of LLMs' solutions by comparing the execution results with ground truth answers. This objective evaluation method is more reliable than pure text-based benchmarks in my perspective.

The authors conduct an ablation study that provides valuable insights into the effectiveness of their benchmark and proposed data synthesis method. This includes examining the impact of removing reasoning steps and filters, as well as comparing forward and reverse data synthesis approaches, along with the visualization of the dataset.

This work fills a gap in existing benchmarks by providing a diverse and challenging dataset for end-to-end optimization problem-solving. Moreover, the proposed ReSocratic method and the resulting RESOCRATIC-29K dataset offer a promising approach to address the data scarcity issue.

**Weaknesses:**

The paper lacks a fine-grained error analysis, which could provide valuable insights into the specific challenges faced by LLMs in optimization tasks. A breakdown of error types, such as errors in understanding the problem, formulating the optimization model, or transferring the mathematical model to code, or summarizing the execution into required formatted outputs, would be helpful to guide future improvements in LLM design and training.

The authors didn't provide an analysis (maybe just a few case studies) of false positives and false negatives in the benchmark results. As with any benchmark, it is expected that both types of errors can exist, and understanding their characteristics could contribute to the benchmark's reliability.

The data collection and annotation process lacks details on deduplication and quality control measures. This helps to ensure the collected optimization problems were diverse enough to measure LLM's modeling capability.

The paper didn't explore the impact of different prompting strategies during inference. For example, investigating whether step-by-step reasoning then generates the code can help improve model performance.

The range of models tested is relatively limited. As a benchmark paper, one would expect a wider variety of LLMs to be included, such as code-specialized models like CodeLLaMA or the Mistral line of models.

The reliability of the back-translation step in the ReSocratic data synthesis is not thoroughly investigated. This helps measure the quality and consistency of the questions generated through back-translation, as well as the potential impact of any errors.

**Questions:**

How does the current perform if using the pass@k metric, which measures the performance of the generated code after k attempts?

The experimental results show that DeepSeek-V2 severely underperforms in the zero-shot setting compared to its few-shot version, while other models do not exhibit such a drastic difference. Is there any reason behind this?

---

> ### Author Response · Authors · 2024-11-23
> **Response to Reviewer hRyJ (1/2)**
>
> Thank you for your time and effort in reviewing our paper. We very much appreciate your insightful comments and your recognition of our work. We hereby address the concerns below.
>
> ### **1. Fine-Grained Error Analysis**
>
> Thank you for pointing out that fine-grained error analysis is important.
> We follow your advice and provide the statistical results on the Code Pass rate under various data categories within the limited time available during the rebuttal period.
> Other error analyses (understanding issues, formulating optimization models, and converting mathematical models into code) require manual annotation, thus they are time-consuming. We will add more analysis in the future.
>
> The results of Code Pass rate are shown as follows ('Code Pass' refers to the success rate of code execution):
>
> | Models | Prompt |  Linear w/o Table | Linear w/ Table | Nonlinear w/o Table | Nonlinear w/ Table | Overall Pass |
> | --- | --- | :---: | :---: | :---: | :---: |  :---:  |
> | ***zero-shot setting*** |
> | Mistral-7B-Instruct-v0.3 | zero-shot |5.8| 6.3| 9.0| 0.1| 6.9|
> | Qwen2-7b-Instruct | zero-shot |21.3| 0.1| 21.8| 12.0| 19.2 |
> | gpt-4o | zero-shot | 93.6 | 93.8 | 83.5 | 78.0 | 90.1 |
> | ***few-shot setting*** |
> | Mistral-7B-Instruct-v0.3 | few-shot |87.7| 85.0| 75.9| 84.0| 83.8|
> | Qwen2-7b-Instruct | few-shot | 92.4| 85.0| 78.9| 82.0| 87.6|
> | gpt-4o | few-shot | 94.2 | 80.0 |  85.7 | 86.0 | 91.7 |
>
> From the above statistical results, we can observe that the Code Pass rate for nonlinear problems is lower than that for linear problems, indicating that solving nonlinear problems poses certain challenges in terms of coding.
> Furthermore, we provide 2 case studies to show the errors in code generation by language models, as shown in the attached comment.
>
> We add the experimental results to Table 6 of our revision.
>
>
> ### **2. Details of Data Collection**
>
> Thank you for pointing out the lack of details regarding Data Collection in our paper.
> We supplement the details as follows:
>
> **Diversity**:
> * Different Data Sources: We assign annotators to collect questions from several textbooks [1][2][3], assignments, and examinations from a university.
> * Data Type Diversity: Annotators are required to collect diverse question types (linear, nonlinear, with table, without table).
> * Deduplication: The annotators are required to write the questions in markdown format. We filter out the questions with identical text.
>
> **Quality Control**:
> * We assure data quality through annotation consistency. Specifically, each question is annotated by 4 annotators, and each annotator performs the annotation independently. Specifically, the annotators are required to write the mathematical models in markdown format for those collected questions.
> * For each question, we assign two other annotators to compare the code execution results of the assigned 4 annotators. If the results are the same, the question is deemed as correctly labeled; otherwise, we will re-label and determine its final result.
>
> We supplement these details in Appendix D.1 of our revision.
>
> ***References***
> [1] Dimitris Bertsimas and John N Tsitsiklis. Introduction to linear optimization, volume 6. Athena
> Scientific Belmont, MA, 1997.
> [2] Michele C, Gerard C, Giacomo Z, et al. Integer programming models. Springer, 2014.
> [3] Laurence A Wolsey. Integer programming. John Wiley & Sons, 2020.
>
>
> ### **3. Impact of Different Prompting Strategies**
>
> Thank you for pointing out the importance of prompting strategies. We follow your advice and construct a new prompt (first step reason then write code) named 'few-shot (first reason)'. The experimental results are shown in the following:
>
> | Models | Prompt | Overall Acc |
> | --- | --- | --- |
> | GPT-3.5-Turbo | few-shot (ori) | 56.4 |
> | GPT-3.5-Turbo | few-shot (first reason) | 55.5 (-0.9) |
> | GPT-4 | few-shot (ori) | 65.5 |
> | GPT-4 | few-shot (first reason) | 63.8 (-1.7) |
> | GPT-4o-mini | few-shot (ori) | 55.0 |
> | GPT-4o-mini | few-shot (first reason) | 54.9 (-0.1) |
> | GPT-4o | few-shot (ori) | 69.4|
> | GPT-4o | few-shot (first reason) | 67.1 (-2.3) |
>
> 'few-shot (ori)' is the original few-shot prompt in out paper. It can be seen from the results that "first understanding the reasoning and then writing code" does not significantly improve performance.
>
> We add these experimental results to Table 9 of our revision. The 'few-shot (first reason)' prompt is provided in Appendix E.1.3 of our revision.

---

> > ### Author Response · Authors · 2024-11-23
> > **Error Case Studies**
> >
> > #### **Error Case 1**
> > ```python
> > import pyscipopt
> >
> > # Create a new model
> > model = pyscipopt.Model()
> >
> > # Define variables
> > acai_smoothies = model.addVar("acai_smoothies", vtype="I", lb=0)
> > banana_smoothies = model.addVar("banana_smoothies", vtype="I", lb=0)
> >
> > # Define objective function
> > # Minimize the total amount of water used
> > obj = model.addVar('obj')
> > model.setObjective(obj, "minimize")
> > model.addCons(obj == 3 * acai_smoothies + 4 * banana_smoothies)
> >
> > # Add constraints
> > # Constraint for acai berries
> > model.addCons(7 * acai_smoothies <= 3500)
> >
> > # Constraint for banana chocolate
> > model.addCons(6 * banana_smoothies <= 3200)
> >
> > # Constraint for more banana smoothies than acai smoothies
> > model.addCons(banana_smoothies > acai_smoothies)
> >
> > # Constraint for at least 35% of smoothies being acai berry smoothies
> > model.addCons(acai_smoothies >= 0.35 * (acai_smoothies + banana_smoothies))
> >
> > # Solve the problem
> > model.optimize()
> > ```
> > #### **Error Info**
> > ```
> > NotImplementedError: Can only support constraints with '<=', '>=', or '=='.
> > ```
> > PySCIPOpt does not support the symbols <, >; it only supports <=, >=.
> >
> >
> > #### **Error Case 2**
> > ```python
> > import math
> > import pyscipopt
> >
> > # Create a new model
> > model = pyscipopt.Model()
> >
> > # Define variables for the rescue team's location (x, y)
> > x = model.addVar('x', lb=0)  # x must be non-negative
> > y = model.addVar('y')
> >
> > # Define the man's location
> > man_x = 3/4
> > man_y = 0
> >
> > # Define the objective function: minimize the distance to the man
> > distance = model.addVar('distance')
> > model.setObjective(distance, "minimize")  # We want to minimize the distance
> >
> > # Add the distance constraint
> > model.addCons(distance == math.sqrt((x - man_x)**2 + (y - man_y)**2))
> >
> > # Add the constraint for the swamp: y >= x^2
> > model.addCons(y >= x**2)
> >
> > # Solve the problem
> > model.optimize()
> > ```
> >
> > #### **Error Info**
> > ```
> >     model.addCons(distance == math.sqrt((x - man_x)**2 + (y - man_y)**2))
> > TypeError: must be real number, not pyscipopt.scip.Expr
> > ```
> > In PySCIPOpt, you cannot use math.sqrt() directly.

---

> ### Author Response · Authors · 2024-11-23
> **Response to Reviewer hRyJ (2/2)**
>
> ### **4. The Range of Models Tested is Relatively Limited**
>
> We add a bunch of models as baselines following your suggestion. We will continue to maintain the leaderboard of this benchmark and will include more models to guide the research community.
>
> | Models | Prompt | Linear w/o Table | Linear w/ Table | Nonlinear w/o Table | Nonlinear w/ Table | Overall | Code Pass |
> | --- | --- | :---: | :---: | :---: | :---: | :---: | :---: |
> | Mistral-7B-Instruct-v0.3 | zero-shot | 0.6 | 0.0 | 0.0 | 0.0 | 0.3 | 6.9 |
> | Mistral-7B-Instruct-v0.3 | few-shot  | 40.0 | 23.8 | 13.5 | 18.0 | 27.9 | 83.8 |
> | Qwen2-7b-Instruct | zero-shot | 3.5 | 0.0  | 3.0 | 0.0 | 2.6 | 19.2 |
> | Qwen2-7b-Instruct | few-shot | 65.5 | 27.5 | 18.8 | 14.0 | 46.0 | 87.6 |
> | deepseek-v2.5 | zero-shot | 78.4 | 67.5 | 33.1 | 24.0 | 62.5 | 92.7 |
> | deepseek-v2.5 | few-shot  | 79.5 | 71.3 | 40.6 | 48.0 | 67.3 | 91.2 |
> | gpt-4o-mini | zero-shot | 76.0 | 48.8 | 35.3 | 34.0 | 60.0 | 84.8 |
> | gpt-4o-mini | few-shot  | 74.6 | 52.5 | 14.3 | 34.0 | 55.0 | 74.4 |
> | gpt-4o | zero-shot | 78.1 | 65.0 | 45.9 | 40.0 | 66.1 | 90.1 |
> | gpt-4o | few-shot  | 81.0 | 63.8 | 50.4 | 50.0 | 69.4 | 91.7 |
>
> These experimental results have been added to Table 2 in our manuscript.
>
>
> ### **5. Reliability of Back-Translation Step in ReSocratic**
> The back-translation step is a simple task, we show an example in the attached comment.
> During the rebuttal period, we asked 3 master's degree students in computer science to examine 20 back-translation samples to determine whether the generated questions in this step are correct. Specifically, the following points were checked:
> * Whether the generated question text includes all the information from the demonstration.
> * Whether the generated question is consistent with the question in the demonstration.
>
> The test results show that all 20 samples are correct.
>
>
> ### **6. Pass@k Performance**
> We follow your advice and provide the Pass@k results as follows (we set the temperature as 0.7):
>
> | Models | Pass@5 | Pass@10 | Pass@15 | Pass@20 | Pass@25 | Pass@30 |
> | --- | :---: | :---: | :---: | :---: | :---: | :---: |
> | Llama-3-8B-Instruct | 40.5 | 55.5 | 59.5 | 61.5 | 62.5 | 63.3 |
> | Mistral-7B-Instruct-v0.3 | 47.1 | 57.0 | 61.3 | 63.8 | 64.7 | 66.1 |
> | Qwen2-7b-Instruct | 59.5 | 63.5 | 65.8 | 67.3 | 68.3 | 68.6 |
>
> The performance of the model improves with the increase in the number of generation attempts.
> We add these experimental results in Table 10 of our revision.
>
> ### **7. DeepSeek-V2 severely underperforms in the zero-shot setting**
>
> The experimental results we added above show that similar phenomena occur in both Mistral-7B-Instruct-v0.3 and Qwen2-7b-Instruct. Moreover, unlike DeepSeek-V2, DeepSeek-V2.5 (the combination of DeepSeek-V2-Chat and DeepSeek-Coder-V2) does not exhibit a significant performance gap in zero-shot and few-shot settings. We believe that this phenomenon is caused by the drawbacks of the model's coding capabilities. As can be seen from the "Code Pass" indicator, models with weaker coding abilities are unable to correctly write code relying on their own capabilities in a zero-shot setting but can solve problems correctly when provided with few-shot examples.

---

> > ### Author Response · Authors · 2024-11-23
> > **An example of back-translation**
> >
> > **Demonstration**:
> > ```
> > ## Define Variables:
> > A cereal company makes nutritional cereal, kids' cereal, and sugary cereal. The company needs to determine the optimal number of boxes to produce for each type of cereal.
> > // {"number of nutritional cereal boxes": "x", "range": "x >= 0", "type": "integer"}
> > // {"number of kids' cereal boxes": "y", "range": "y >= 0", "type": "integer"}
> > // {"number of sugary cereal boxes": "z", "range": "z >= 0", "type": "integer"}
> >
> > ## Define Objective Function:
> > The revenue per box of nutritional cereal is $1, the revenue per kids' cereal is $1.50, and the revenue per sugary cereal is $2. How many of each should they make to maximize revenue?
> > // Maximize x + 1.5y + 2z
> >
> > ## Generate Constraint-1:
> > Each box of nutritional cereal requires 3 units of oat, each kids' cereal requires 1.5 units of oat, and each sugary cereal requires 2 units of oat. The company has available 500 units of oat.
> > // 3x + 1.5y + 2z <= 500
> >
> > ## Generate Constraint-2:
> > Each box of nutritional cereal requires 1 unit of sugar, each kids' cereal requires 1.5 units of sugar, and each sugary cereal requires 4 units of sugar. The company has available 700 units of sugar.
> > // x + 1.5y + 4z <= 700
> > ```
> >
> > **Back-translated question**:
> > ```
> > A cereal company makes nutritional cereal, kids' cereal, and sugary cereal. The company needs to determine the optimal number of boxes to produce for each type of cereal. Each box of nutritional cereal requires 3 units of oat, each kids' cereal requires 1.5 units of oat, and each sugary cereal requires 2 units of oat. The company has available 500 units of oat. Each box of nutritional cereal requires 1 unit of sugar, each kids' cereal requires 1.5 units of sugar, and each sugary cereal requires 4 units of sugar. The company has available 700 units of sugar. The revenue per box of nutritional cereal is $1, the revenue per kids' cereal is $1.50, and the revenue per sugary cereal is $2. How many of each should they make to maximize revenue?
> > ```
> >
> > As shown in this example, back-translation is almost equivalent to extracting the natural language part of the demonstration, which is a very simple task. We add the above sample in the Appendix.

---

> > ### Comment · Reviewer_hRyJ · 2024-11-25
> >
> > I want to thank the authors for the additional experiments! They did address most of my concern. However, my primary expectation is the fine-grained error analysis (other error analyses as the authors mentioned in the rebuttal), which should play an important role in any benchmarking work. Hence, I maintained my original score.

---

> > > ### Author Response · Authors · 2024-11-29
> > >
> > > Dear Reviewer hRyJ,
> > >
> > > We sincerely appreciate your thorough review and valuable feedback. We fully understand you might be quite busy. However, as the discussion deadline is approaching, would you mind checking our follow-up additional response about fine-grained error analysis? We would like to know if our recent efforts have addressed your remaining concerns. We are fully committed to making any necessary revisions and welcome any further comments or discussions you may have.
> > >
> > > Thank you once again for your time and effort in reviewing our paper.
> > >
> > > Best Regards,
> > >
> > > The authors of Submission 8641

---

> > > > ### Comment · Reviewer_hRyJ · 2024-12-02
> > > >
> > > > I would like to thank the authors for the additional `LLM as a judge` evaluation. The fine-grained error analysis I was expecting should have more categories than the provided two. However, the additional result provides new insights to understand the problem. I will increase my score to 8.

---

> > > > > ### Author Response · Authors · 2024-12-03
> > > > >
> > > > > Dear Reviewer hRyJ,
> > > > >
> > > > > Thank you very much for your positive feedback and for considering our additional response. We're truly grateful for your insights.
> > > > >
> > > > > We also appreciate your decision to increase our score to 8. We noticed that the updated score hasn't appeared yet, and we would be grateful if you could take a moment to look into this when convenient.
> > > > >
> > > > > Thank you once again for your support and guidance.
> > > > >
> > > > > Best,
> > > > > The authors of Submission 8641

---

> > > > > ### Author Response · Authors · 2024-12-04
> > > > > **Gratitude**
> > > > >
> > > > > Dear Reviewer hRyJ,
> > > > >
> > > > > I would like to express our sincere gratitude for taking the time to update the rating. Your dedication to the review process is greatly appreciated.
> > > > >
> > > > > Best, The authors of Submission 8641

---

> ### Author Response · Authors · 2024-11-26
> **Additional Response to Reviewer hRyJ About Error Analysis**
>
> Thank you again for your feedback. We are very pleased to know that our additional experiments address most of your concerns. Following your advice, we have been delving deeper into error analysis. Finally, through our relentless efforts, we are able to measure the *accuracy of the mathematical formulations* and the *accuracy of transferring the mathematical formulations to code* using the "**LLM as a judge**" approach. Moreover, we have conducted human-LLM consistency statistics to confirm our results. This analysis can provide a more comprehensive understanding of the models' performance.
>
> We sincerely hope that the newly added *accuracy of the mathematical formulations* and *accuracy of transferring the mathematical formulations to code* together with the *code pass rate* mentioned in our previous response can address your concern.
>
> In our experiment, we use deepseek-v2.5 to judge the results. The experimental results are shown in the following additional tables.
>
> **Additional Table 1: Accuracy of the Mathematical Formulations.**
>
> | Models | Prompt |  Linear w/o Table | Linear w/ Table | Nonlinear w/o Table | Nonlinear w/ Table | Overall |
> | --- | --- | :---: | :---: | :---: | :---: |  :---:  |
> | Mistral-7B-Instruct-v0.3 | zero-shot | 56.4 | 25.0 | 23.3 | 24.0 | 42.3 |
> | Mistral-7B-Instruct-v0.3 | few-shot | 70.7 | 38.8 | 32.3 | 46.0 | 56.0 |
> | Qwen2-7b-Instruct | zero-shot | 71.1 | 40.0 | 39.1 | 26.0 | 56.2 |
> | Qwen2-7b-Instruct | few-shot  | 78.7 | 43.8 | 41.4 |46.0 | 63.1 |
> | deepseek-v2.5 | zero-shot | 91.8 | 77.5 | 63.2 | 56.0 | 80.7 |
> | deepseek-v2.5 | few-shot  | 93.6 | 81.3 | 70.7 | 68.0 | 84.8 |
> | gpt-4o-mini | zero-shot | 89.5 | 72.5 | 69.2 | 60.0 | 80.3 |
> | gpt-4o-mini | few-shot  | 90.1 | 67.5 | 72.9 | 70.0 | 81.7 |
> | gpt-4o | zero-shot | 90.6 | 82.5 | 80.5 | 74.0 | 85.9 |
> | gpt-4o | few-shot  | 92.9 | 73.8 | 82.0 | 70.0 | 86.1 |
>
>
> As shown in the table, small open-source models like Mistral-7B-Instruct-v0.3 and Qwen2-7b-Instruct tend to make more mistakes in mathematical formulations than large models like deepseek-v2.5 and gpt-4o. Moreover, the few-shot setting can significantly decrease the errors of mathematical formulations for all models than the zero-shot setting. These phenomena are also observed in benchmarks like GSM8K and MATH, leading us to conclude that they stem from the model's capacity for mathematical reasoning.
>
>
> **Additional Table 2: Accuracy of Transferring the Mathematical Formulations to Code.**
>
> | Models | Prompt |  Linear w/o Table | Linear w/ Table | Nonlinear w/o Table | Nonlinear w/ Table | Overall |
> | --- | --- | :---: | :---: | :---: | :---: |  :---:  |
> | Mistral-7B-Instruct-v0.3 | zero-shot | 1.0 | 0.0 | 0.0 | 0.0 | 0.7 |
> | Mistral-7B-Instruct-v0.3 | few-shot | 50.8 | 61.3 | 41.9 | 39.1 | 49.9 |
> | Qwen2-7b-Instruct | zero-shot | 4.9 | 0.0 | 7.7 | 0.0 | 4.7 |
> | Qwen2-7b-Instruct | few-shot  | 83.3 | 62.9 | 45.5 | 30.4 | 72.8 |
> | deepseek-v2.5 | zero-shot | 85.4 | 87.1 | 52.4 | 42.9 | 77.5 |
> | deepseek-v2.5 | few-shot  | 85.0 | 87.7 | 57.4 | 70.6 | 79.3 |
> | gpt-4o-mini | zero-shot | 85.0 | 67.2 | 51.1 | 56.7 | 74.4 |
> | gpt-4o-mini | few-shot  | 82.8 | 77.8 | 19.6 | 48.6 | 67.4 |
> | gpt-4o | zero-shot | 86.1 | 78.8 | 56.0 | 57.0 | 76.9 |
> | gpt-4o | few-shot  | 87.1 | 86.4 | 61.5 | 71.4 | 80.6 |
>
> For models gpt-4o, gpt-4o-mini, and deepseek-v2.5, their errors in mathematical formulations are relatively few. In particular, gpt-4o achieves the SOTA performance under both prompt settings. Moreover, as can be seen from Additional Table 2, the code transfer accuracy of Mistral-7B-Instruct-v0.3 and Qwen2-7b-Instruct is significantly lower in the zero-shot setting compared to the few-shot setting, which echoes point 7 of our previous response (*models with weaker coding abilities are unable to correctly write code relying on their capabilities in a zero-shot setting but can solve problems correctly when provided with few-shot examples*).
>
> We add these experimental results to Table 7 and Table 8 of our revision. The judge prompt is shown in Appendix E.1.5.
>
>
> **Human-LLM consistency Statistics**
>
> In addition, we have conducted human-LLM consistency statistics. We randomly select 20 samples and ask 2 master's degree students in computer science to label them, comparing the labels to those of "LLM as a judge". The final results show that 19 out of 20 samples were consistent. This indicates the effectiveness of "LLM as a judge" in fine-grained error analysis.

---

> ### Author Response · Authors · 2024-12-04
> **Kindly Remind of Rating Update**
>
> Dear Reviewer hRyJ,
>
> As the discussion deadline is approaching, we noticed that your promised score increase from 6 to 8 hasn't appeared yet. We would be grateful if you could take a moment to look into this when convenient.
>
> Best, The authors of Submission 8641

---

### Author Response · Authors · 2024-12-03
**Global Response**

We thank the area chair and all reviewers for your time, insightful suggestions, and valuable comments. Your suggestions have been invaluable in refining our work, and we deeply appreciate the time and effort you dedicated to reviewing our paper. We have carefully addressed all points and incorporated the necessary improvements in the revised version.

We are pleased that reviewers appreciated:
* The diverse and challenging benchmark that effectively evaluates the optimization problem-solving abilities. (*hRyJ*, *GR55*)
* The novelty of our data synthesis method. (*GR55*, *PMkM*)
* The clarity illustration of the proposed benchmark and method. (*PMkM*)


We also sincerely thank the reviewers for their valuable suggestions, which helped us identify areas for improvement. In response to their feedback, we have made several major revisions and additions:

* **Fine-Grained Error Analysis** (*hRyJ*, *GR55*): We follow the reviewers' advice and provide the statistical results on the Code Pass rate under various data categories. Moreover, we evaluate the *accuracy of the mathematical formulations* and the *accuracy of transferring the mathematical formulations to code* using the "LLM as a judge" approach. (Appendix B.1)
* **Expand the scope of tested LLMs** (*hRyJ*): We add a bunch of models as baselines following the reviewer's suggestion. We will continue to maintain the leaderboard of this benchmark and will include more models to guide the research community. (Table 2)
* **Details of Data Collection** (*hRyJ*, *PMkM*): We add a detailed description of the data collection process in the main paper. (Section 3 & Appendix D.1)
* **More Detailed Analysis** (*hRyJ*, *GR55*): We follow the reviewers' advice and provide more detailed analysis, including *Impact of Different Prompting Strategies*, *Pass@k Performance*, and *Solving Efficiency of the Generated Codes*. (Appendix B.2 - B.4)


We are pleased that reviewer *hRyJ* and reviewer *GR55* both indicated that our response has addressed most of their concerns. We thank all the reviewers again for their time and effort in reviewing our paper and are committed to addressing every issue raised.

Sincerely,

The authors of Submission 8641

---

### Meta-Review · Area_Chair_8gGZ · 2024-12-21

**Metareview:**

The paper presents a benchmark for evaluating LLMs on optimization problems, along with ReSocratic, a data synthesis method. The work includes a comprehensive benchmark dataset and demonstrates improvements in open-source LLM performance through fine-tuning. Overall, the work represents a solid contribution to the field, with clear practical value and well-documented methodology, despite some limitations in scope and novelty.

**Additional Comments On Reviewer Discussion:**

During the review process, the authors provided substantial additional data, including a detailed "LLM as judge" error analysis system with human-validation consistency checks, expanded model testing results, and clear differentiation from existing benchmarks. The discussion process was productive, leading one reviewer to increase their score from 6 to 8, while two other reviewers maintained scores of 6, acknowledging the improvements while retaining some concerns about scalability to larger problems and overlap with existing methods.

---

### Decision · Program_Chairs · 2025-01-22

Accept (Poster)